# Effects of the terdiurnal tide on the Sporadic-E layers (*Es*) development at low latitudes over the Brazilian sector

Pedro A. Fontes[1,2*], Marcio T. A. H. Muella[1], Laysa C. A. Resende[3,4], Vânia F. Andrioli[3,4], Paulo R. Fagundes[1], Valdir G. Pillat[1], Paulo P. Batista[3], Alexander J. Carrasco[3,5]

[1]Universidade do Vale do Paraíba – Univap, São José dos Campos – SP, Brasil.

[2]Instituto Federal do Maranhão – IFMA, Bacabal – MA, Brasil.

[3]Instituto Nacional de Pesquisas Espaciais – INPE, São José dos Campos-SP, Brasil.

[4]State Key Laboratory of Space Weather, Beijing, China.

[5]Universidad de Los Andes, Mérida, Venezuela.

*Corresponding author*: Pedro Fontes (pedro.fontes@ifma.edu.br; pedro.fontes2@hotmail.com)

**Abstract.** Sporadic-E (*Es*) layers are patches of high ionization observed at around 100-140 km height in the E region. Their formation at low latitudes primarily associated with the diurnal and semidiurnal components of the tidal winds via the ion convergence driven by the wind shear mechanism. However, recent studies have shown the influence of other tidal modes, such as the terdiurnal tide. Therefore, this work investigates the effect of terdiurnal tide-like oscillations on the
occurrence/formation of the *Es* layers observed over Palmas (10.17° S; 48.33° W; dip lat. -7.31°), a low latitude station in Brazil. The analysis was conducted from December/2008 to November/2009 by using data collected from a CADI (Canadian Advanced Digital Ionosonde) ionosonde. Additionally, the E Region Ionospheric Model (MIRE) was used to simulate the terdiurnal tidal component in the *Es* layer development. The results show modulations of 8 hours periods on the occurrence rates of the *Es* layers during all seasonal periods. In general, we see three well-defined peaks in a superimposed summation of
the *Es* layer types per hour in summer and autumn. We also observed that the amplitude modulation of the terdiurnal tide on the *Es* occurrence rates minimizes in December in comparison to the other months of the summer season. Other relevant aspects of the observations, with complementary statistical and periodogram analysis, are highlighted and discussed.

## 1 Introduction

Sporadic-E layers (*Es*) are thin and dense layers observed in the ionospheric E region at altitudes ranging from 100 to 140 km.
The ion density of the *Es* layer is associated with molecular ions, mostly $O_2^+$ and $NO^+$, and the main metal ions, $Fe^+$, $Mg^+$, and $Na^+$. Metal ions have a longer lifetime than the molecular ions, making the *Es* layers long-lasting (Whitehead, 1989; Plane, 2003).

The *Es* layers are divided into three different classes: equatorial layers, low- and mid-latitude layers, and the auroral or high-latitude layers. They are differentiated into types designated by the lowercase letters of the alphabet, such as $Es_c$ (cusp), $Es_f$
(flat), $Es_h$ (high), $Es_l$ (low), $Es_q$ (equatorial), $Es_a$ (auroral), and $Es_s$ (slant). The '*c*', '*h*', '*l*', and '*q*' types are formed in the

daytime, while the '*f*' type is usually nocturnal. Type '*h*' of *Es* occurs at higher altitudes, generally ranging between 120 and 140 km, and can appear and disappear suddenly, as well as evolving into type '*c*' at lower altitudes (Conceição-Santos et al., 2020). The '*l*' and '*f*' types generally occur at 100-110 km. Furthermore, the '*q*' type occurs near the magnetic equator around 100 km (Resende et al., 2017b; Conceição-Santos et al., 2020).

The *Es* layers occur at most latitudes and longitudes and have different mechanisms of formation that depend mainly on magnetic latitude (Kirkwood and Nilsson, 2000). The vertical shear in the east-west component of horizontal winds (wind shear) is the most accepted theory to explain the formation of *Es* layers. According to this theory, a vertical convergence of the charged particles occurs by $U \times B$ forces, where $U$ and $B$ denote the horizontal wind and the Earth's geomagnetic field, respectively (Whitehead, 1961; 1989; Kirkwood and Nilsson, 2000; Haldoupis, 2012). The atmospheric tides are one of the

possible sources of wind shears, which means that solar tides can strongly control the intensification of *Es* layers (Forbes and Wu, 2006; Forbes et al., 2008; Moudden and Forbes, 2013). Solar tides are oscillations in the atmosphere that occur due to the absorption of infrared and EUV radiation by water ($H_2O$), ozone ($O_3$), and molecular oxygen ($O_2$) in the troposphere, stratosphere, and lower thermosphere. This absorption of radiation causes periodic heating and expansion of tidal amplitudes that grow exponentially in height as they are excited into regions of the lower thermosphere between the altitudes of 90 and

120 km (Forbes and Wu, 2006; Forbes et al., 2008; Smith, 2012; Pancheva et al., 2013).

The source of terdiurnal tide generation has two main discussions currently in the literature, one being the nonlinear interaction between the diurnal and semidiurnal tides (Teitelbaum and Vial, 1991; Huang et al., 2007; Forbes and Wu, 2006; Forbes et al., 2008; Truskowski et al., 2014; Forbes et al., 2014), and the other is direct solar heating (Akmaev, 2001; Smith and Ortland, 2001; Du and Ward, 2010; Lilienthal and Jacobi, 2019). For Truskowski et al. (2014), the terdiurnal tide is generated by the

nonlinear interaction between the diurnal and semidiurnal tides at mid- and low-latitude, but they did not rule out the contribution of direct warming. Lilienthal and Jacobi (2019) showed that direct heating is the primary source of terdiurnal tide generation at mid- and low latitude in the northern and southern hemispheres, with nonlinear interaction being a secondary source. The seasonality of the terdiurnal tidal amplitude has been studied in the interest of understanding its source of generation and its variability in the lower thermosphere, which can have maxima in different months and latitudes of the

hemispheres (Zhao et al., 2005; Jiang et al., 2009; Venkateswara Rao et al., 2011; Moudden and Forbes, 2013; Fytterer et al., 2014; Jacobi and Arras, 2019).

The phase and vertical wavelength of the terdiurnal tide vary as a function of season and latitude and differ between the zonal and meridional components (Zhao et al., 2005; Venkateswara Rao et al., 2011; Jacobi and Arras, 2019). Jacobi and Arras (2019) compared data results between the Collm (51° N) and Obninsk (55° N) meteoric radars, showing the variability of tidal

amplitudes between seasons, but the annual seasonal cycles were approximately the same for the zonal and meridional component at these latitudes. Moudden and Forbes (2013) also found periodic seasonality of the terdiurnal tide at mid- and low latitudes when they focused their analyses on the TW3, TE1, TW4, and TW5 (Terdiurnal tides for east (TE) and west (TW) with zonal numbers 1, 3, 4 and 5) to try answering why these tides have well-defined and repeatable seasonal behaviors from year to year in the northern and southern hemispheres. The authors pointed out that the TW3 tide has the greatest

amplitude near the equator in both hemispheres and has two distinct peaks that occur regularly each year above 110 km on the autumn and spring equinoxes and in the month of February. According to the authors, this tide showed similar characteristics to diurnal tide (DW1) and semidiurnal tide (SW2) with respect to annual periodicity and seasonal variability of amplitude in the latitudes of the two hemispheres. Fytterer et al. (2014) analyzed the effect of the terdiurnal tide on the frequencies (*foEs*) of *Es* layers using Radio Occultation (RO) data over a global distribution between ±60°, revealing nearly uniform numbers of

*Es* layer occurrences and terdiurnal tide amplitude across years at latitudes in both hemispheres. The authors used the months of January and April, respectively, as representative of the summer solstice and autumn equinox, observing good agreement between the terdiurnal amplitude and the occurrences of *Es* layers with 8-h shear characteristics. Jacobi and Arras (2019) analyzed the diurnal, semidiurnal, and terdiurnal tidal phases over the *Es* layers using one year as representative of a seasonal tidal cycle. The authors concluded that the tides observed in the *Es* layers are largely due to neutral zonal wind shear in the

presence of a component of the Earth's magnetic field. Tidal winds maintain a certain annual periodicity across latitudes (Moudden and Forbes, 2013; Fytterer et al., 2014; Jacobi et al., 2017; Jacobi and Arras, 2019) so that a one-month analysis can represent the tide in a season and a one-year analysis can represent a tidal cycle at a given latitude in the hemispheres. It is worth mentioning that this work is restricted to presenting a possible effect of the terdiurnal tide on the *Es* layer formation and it is not concerned to the behavior of the terdiurnal tide.

The type of atmospheric tide can influence the development of the sporadic layers (Haldoupis and Pancheva, 2006; Haldoupis, 2012; Lilienthal et al., 2018). The convergence of the *Es* layer to altitudes whose vertical velocities of neutral winds correspond to zero may be driven by the global tidal wind system in the thermosphere (Haldoupis, 2012). The formation and variability of the *Es* layers may be related to the action of atmospheric waves in the lower thermosphere (McLandress, 2002a; 2002b; Haldoupis and Pancheva, 2006; Haldoupis, 2012). Haldoupis et al. (2004) used a time series of *Es* layer parameters scaled

from ionosonde data and found that the dominant amplitudes at the 3 to 36-hour period are the 24, 12, and 8-hour peaks associated, respectively, with the diurnal, semidiurnal, and terdiurnal atmospheric tides. Although the diurnal and semidiurnal tides play leading roles in the occurrence, altitude descent, and strength of *Es* layers, Haldoupis and Pancheva (2006) stressed that the terdiurnal tide can also have a contribution to the convergent shearing of the horizontal winds. In the mid-latitude station of Cyprus (35.0°N; 33.0°E), Oikonomou et al. (2014) observed the presence of terdiurnal tidal effects on *Es* layer that

prevailed only during summer months.

Additionally, Fytterer et al. (2014) used an atmospheric circulation model to support the observations of GPS radio occultation data, and the authors reported a well-pronounced peak in the *Es* layer occurrence at around 10° in both hemispheres associated with the migrating terdiurnal tide. In the Brazilian sector, Resende et al. (2016) and Resende et al. (2017a; 2017b) simulated the *Es* layer using a model called MIRE (E Region Ionospheric Model). They showed that the diurnal and semidiurnal

components are the main ones for forming the *Es* layer in this sector. However, they suggested that some discrepancies between the model and observational data might be due to the exclusion in the analysis of other tidal components such as terdiurnal and quarterdiurnal tides. It is worth mentioning that there are few studies in the literature concerning the role of the terdiurnal tidal

modulation in the formation of *Es* layers, especially in equatorial and low-latitude regions (Haldoupis and Pancheva, 2006; Fytterer et al., 2014; Lilienthal et al., 2018).

Although our knowledge of tidal effects on *Es* layers has advanced over the last 20 years, there are still questions about the possible role of the terdiurnal tide on the formation of *Es* layers at low latitudes. Therefore, the present study analyzes the *Es* layer parameters obtained from ionosonde data from an instrument installed in the low latitude station of Palmas, Brazil (10.17° S; 48.33° W; dip lat.-7.31), to investigate the modulation of the occurrence and formation of *Es* layers associated to the terdiurnal solar tide. In this context, the minimum virtual height (*h'Es*), and the top frequency (*ftEs*) parameters scaled from

ionosonde data were used in this analysis. Moreover, the types of the *Es* layers registered in the ionograms were classified according to the criteria established by the Union of Radio Science (Piggott and Rawer, 1972). During the analyzed period, it was possible to identify four distinct types of *Es* layers in the ionograms recorded by the ionosonde of Palmas. Finally, the E Region Ionospheric Model (MIRE) was used to simulate the effect of terdiurnal (8-h) tidal periodicities on the formation of the *Es* layers.

## 2 Methodology

### 2.1 Data Analysis

The ionosonde used in this work is the CADI (Canadian Advanced Digital Ionosonde) installed in Palmas, Brazil. This ionosonde type operates using a double delta antenna that serves both to transmit and receive the signal. After being processed, the received signal will register the *Es* traces in the ionograms. The transmitter used by the CADI system performs a frequency sweep in the HF range from 1 to 20 MHz with a power of 600 W and a pulse width of 40 s, which gives a height resolution of ±3 km (Gao and MacDougall, 1991; Huang and Macdougall, 2005). The ionograms were processed using the computer

software named UDIDA (Univap Digital Ionsonde Data Analysis), which has been employed to visualize the ionograms on a PC screen (Pillat et al., 2013).

The parameters of virtual height (*h'Es*), top frequency (*ftEs*), blanketing frequency (*fbEs*), and type of the *Es* layers from the ionograms recorded every 5 min were scaled and classified. The different *Es* layer types are defined according to their traces in ionograms, meaning which the physical formation mechanism is acting. Therefore, the *Es* layer classification is given by

lowercase letters following the criteria available in the U.R.S.I. Handbook of Ionogram Interpretation and Reduction (Piggott and Rawer, 1972). We called the *ftEs* since we are not distinguishing between ordinary and extraordinary traces in the data. This parameter refers to the *foEs* in ionosonde data and is related to the maximum frequency that the *Es* layer reaches. We used the months of December of 2008 and January-February of 2009 as representative of summer; the months of March, April, and May of 2009 as representative of autumn; the months of June, July, and August of 2009 as representative of winter; and

finally, the months of September, October, and November of 2009 as representative of spring.

After the *Es* layer reductions with the UDIDA software, the data were hourly adjusted for the three-month periods of summer, autumn, winter, and spring, and also separately for each month of the four seasons. As mentioned before, the ionograms from

UDIDA correspond to 5 minutes of observation. Therefore, in 1 hour of observation, 12 ionograms are obtained, which corresponds to 288 ionograms per day. Considering an ideal (100%) situation of three months (~90 days) for a season, one can observe an overlapping of periods with a total of 1080 (12x90) ionograms in 1 hour. In the calculation of the percentage of occurrence of the *Es* layer types ($Es_{f/l}$, $Es_c$, and $Es_h$), the data set was normalized by subtracting the days with no records. Thus, the percentage of occurrence was calculated through the ratio between the hourly observations of each type of *Es* layer in relation to the total number of measurements recorded at each specific hour throughout the season/month.

To show the peaks of the tidal periodicities, we used an analysis of *fbEs* with the Lomb-Scargle periodogram method (Lomb, 1976; Scargle, 1982). This method is suitable for detecting and characterizing periodic signals in non-uniform sample data, as presented in Vanderplas (2018) and Tacza et al. (2022). In this analysis an adjustment of the minimum frequency to zero was performed, as this minimum does not add computational load and is unlikely to add any significant or artificial peaks to the periodograms (Vanderplas, 2018). In order not to lose relevant information from the data, the periodograms were calculated up to a maximum frequency well-grounded in the ionogram data of the *fbEs*. The *fbEs* values from the four seasons in the year 2008/09 were used in this analysis because they best represent the densities of the *Es* layers.

## 2.2 The MIRE Model

The extended version of the E Region Ionospheric Model (MIRE) was used in this study to calculate the *Es* layers' electronic density profile (Resende et al., 2017a; Conceição-Santos et al., 2019). The MIRE uses the continuity equation to calculate the sum of the ionic density for the main constituents, such as $NO^+, O_2^+, O^+, N_2^+, Fe^+$, and $Mg^+$. The model calculates the time derivative of the ion density, $\partial(N_i)/\partial t$, which depends on the production, $p_i$, the loss, $l_i$, and the transport term, $\partial(V_{iz}[N_i])/\partial z$. Thus, the continuity equations for each ionic species used in the MIRE are given in Equations 1 to 6:

$$[O^+] = \frac{q_{O^+}}{\kappa_1[O_2] + \kappa_2[N_2]}, \tag{1}$$

$$[N_2^+] = \frac{q_{N_2^+}}{\kappa_3[O] + \kappa_4[O_2]}, \tag{2}$$

$$\frac{\partial[O_2^+]}{\partial t} = q_{O_2^+} + \kappa_1[O^+][O_2] + \kappa_4[N_2^+][O_2] - \kappa_5[O_2^+][NO] - \alpha_{O_2^+}[O_2^+]n_e - \frac{\partial\left(V_{O_2^+}\right)[O_2^+]}{\partial z}, \tag{3}$$

$$\frac{\partial[NO^+]}{\partial t} = q_{NO^+} + \kappa_2[O^+][N_2] + \kappa_3[N_2^+][O] - \kappa_5[O_2^+][NO] - \alpha_{NO^+}[NO^+]n_e - \frac{\partial(V_{NO^+})[NO^+]}{\partial z}, \tag{4}$$

$$\frac{\partial[Fe^+]}{\partial t} = [Fe].(j_1 + [NO^+]\gamma_{12} + [O_2^+]\gamma_{14} + [O^+]\gamma_{15}) - [Fe^+]n_e\gamma_2 - [Fe^+].\{[N_2].([O_2]\gamma_{10} + [N_2]\gamma_{11} + [O]\gamma_{11})\}$$
$$- \frac{\partial(V_{Fe^+})[Fe^+]}{\partial z}, \tag{5}$$

$$\frac{\partial [Mg^+]}{\partial t} = [Mg].\left(j_1' + [NO^+]\gamma_{12}' + [O_2^+]\gamma_{14}' + [O^+]\gamma_{15}'\right) - [Mg^+]n_e\gamma_2'$$
$$- [Mg^+].\{[N_2].([O_2]\gamma_{10}' + [N_2]\gamma_{11}' + [O]\gamma_{11}')\} - \frac{\partial(V_{Mg^+})[Mg^+]}{\partial z}. \tag{6}$$

The chemical reactions and respective rate coefficients for production and loss used in MIRE is shown in Table 1. These coefficients were obtained in Chen and Harris (1971) for the molecular ions, and in Carter and Forbes (1999) for the metallic ions. The coefficients for $Mg^+$ are half that of $Fe^+$. For detailed information about these equations and coefficients, see Carrasco et al. (2007) and Resende et al. (2017a; 2017b).

**Table 1: Chemical reactions for the molecular and metallic constituents used in MIRE.**

| Molecular Production Rate ($cm^{-3}\,s^{-1}$) | |
|---|---|
| $O_2 + h\nu \rightarrow O_2^+ + e$ | $q_{O_2^+}$ |
| $N_2 + h\nu \rightarrow N_2^+ + e$ | $q_{N_2^+}$ |
| $NO + h\nu \rightarrow NO^+ + e$ | $q_{NO^+}$ |
| $O + h\nu \rightarrow O^+ + e$ | $q_{O^+}$ |
| **Rate Coefficient ($cm^{-3}\,s^{-1}$)** | |
| $O^+ + O_2 \rightarrow O_2^+ + O$ | $\kappa_1 = 4.0\ x\ 10^{-11}$ |
| $O^+ + N_2 \rightarrow NO^+ + N$ | $\kappa_2 = 1.3\ x\ 10^{-12}$ |
| $N_2^+ + O \rightarrow NO^+ + N$ | $\kappa_3 = 2.5\ x\ 10^{-10}$ |
| $N_2^+ + O_2 \rightarrow O_2^+ + N_2$ | $\kappa_4 = 1.0\ x\ 10^{-10}$ |
| $O_2^+ + NO \rightarrow NO^+ + O_2$ | $\kappa_5 = 8.0\ x\ 10^{-10}$ |
| $NO^+ + e \rightarrow N + O$ | $\alpha_{NO^+} = 4.7\ x\ 10^{-7}(300/Te)$ |
| $O_2^+ + e \rightarrow O + O$ | $\alpha_{O_2^+} = 2.2\ x\ 10^{-7}(300/Te)^{0.7}$ |
| **Metallic Production Rate** | |
| $Fe + h\nu \rightarrow Fe^+ + e$ | $j_1 = 5.0\ x\ 10^{-7}\ s^{-1}$ |
| $Fe^+ + e \rightarrow Fe + 7.9ev$ | $\gamma_2 = 1.0\ x\ 10^{-12}\ cm^3\ s^{-1}$ |
| $Fe^+ + O_2 + N_2 \rightarrow FeO_2^+ + N_2$ | $\gamma_{10} = 2.5\ x\ 10^{-30}\ cm^6\ s^{-1}$ |
| $Fe^+ + N_2 + N_2 \rightarrow FeN_2^+ + N_2$ | $\gamma_{11} = 2.5\ x\ 10^{-30}\ cm^6\ s^{-1}$ |
| $Fe^+ + O + N_2 \rightarrow FeO^+ + N_2$ | $\gamma_{12} = 2.5\ x\ 10^{-30}\ cm^6\ s^{-1}$ |
| $Fe + NO^+ \rightarrow Fe^+ + NO$ | $\gamma_{13} = 7.0\ x\ 10^{-10}\ cm^3\ s^{-1}$ |
| $Fe + O_2^+ \rightarrow Fe^+ + O_2$ | $\gamma_{14} = 9.4\ x\ 10^{-10}\ cm^3\ s^{-1}$ |
| $Fe + O^+ \rightarrow Fe^+ + O$ | $\gamma_{15} = 2.0\ x\ 10^{-9}\ cm^3\ s^{-1}$ |

Therefore, the electronic density profile is given by the sum of the molecular and metal ions, as shown in Equation 7. The simulations are obtained in a space-time of approximately 0.2 km height every 2 minutes between 00-24 UT for altitudes between 86 and 140 km, assuming photochemical equilibrium of the species as initial conditions for the numerical equations to be solved until ion convergence is reached (Carrasco et al., 2007; Resende et al., 2016; 2017a; 2017b; Conceição-Santos et al., 2019; 2020).

$$n_e = [O_2^+] + [NO^+] + [O^+] + [N_2^+] + [Fe^+] + [Mg^+].$$ (7)

For the transport term in the continuity equation, the MIRE uses the equation of motion that depends on the meridional $(U_x)$ and zonal $(U_y)$ wind components, and the electric field $(E_{x,y,z})$, as given in Equation 8.

$$V_{iz} = \frac{\omega_i^2}{(v_{in}^2 + \omega_i^2)} \left[ cosI.sinI.U_x + \frac{v_{in}}{\omega_i}.cosI.U_y + \frac{1}{v_{in}}\frac{e}{m_i}.cosI.sinI.E_x + \frac{e}{\omega_i m_i}.cosI.E_y \right.$$
$$\left. + \frac{e}{v_{in} m_i}.\left(\frac{v_{in}^2}{\omega_i^2} + sin^2 I\right).E_z \right],$$ (8)

where $\omega_i$ is the ion's gyrofrequency; $v_{in}$ is the ion-neutral collision frequency; $I$ is the magnetic inclination angle; $m_i$ is the ion's mass; and $e$ is the ion's electric charge. The coordinate system is composed of the x-axis pointing south, the y-axis pointing east, and the z-axis pointing up.

In this work, the wind data used in MIRE was obtained by the SkiYMET meteor radar installed in the OLAP observatory at São João do Cariri (7.23º S; 36.32º W; dip lat. -12.21). The SkiYMET radar monitors the meteor echoes for altitudes ranging from 80 to 100 km (Hocking, 2004; Buriti et al., 2008; Andrioli et al., 2009). However, the height range of interest in this study at MIRE is between 86 and 140 km (Resende et al., 2017a; Resende et al., 2017b). Thus, meridional and zonal wind data observed on meteoric radars were extrapolated with a Lorentz curve fitting for heights from 100 to 140 km. The Lorentz curve 160 was used here for wind amplitude, considering the theory about wind behavior (Lindzen and Chapman, 1969; Forbes and Garrett, 1979), and according to equation 9:

$$U_{(x0,y0)}^{(24,12,8)}(z) = \mu_{(x0,y0)} + \frac{2.A_{(x0,y0)}}{\pi}.\frac{\varphi_{(x,y)}}{4.\left(z - h_{(x0,y0)}\right)^2 + \varphi_{(x,y)}^2},$$ (9)

where $\mu_{(x0,y0)}$, $A_{(x0,y0)}$, $\varphi_{(x,y)}$ and $h_{(x0,y0)}$ are the fitted parameters for the meridional and zonal components of the diurnal (24 h), semidiurnal (12 h) and terdiurnal (8 h) tides. Additionally, the wind phases were obtained by a linear fitting of the observational data. The vertical wavelengths $(\lambda_{x,y})$ can be used from the respective wave phase equations by multiplying the 165 linear coefficient of the fitted curves by the corresponding tidal period, e.g., 24 h for diurnal tide, 12 h for the semidiurnal tide, and 8 for the terdiurnal (Buriti et al., 2008). It is important to mention here that we used the extended version of MIRE proposed

by Conceição-Santos et al. (2019) with a modification in the $\lambda_{x,y}$ of the extrapolated winds, keeping them constant between 100 and 120 km, and tending to infinity between 120 and 140 km. Then they were inserted from the MIRE model to show the *Es* layer density profiles for the months of December (summer), April (autumn), July (winter) and October (spring) as representative of the seasons. Finally, the wind profiles agreed with the previous study performed with Wind Image Interferometer (WINDII) aboard the Upper Atmosphere Research Satellite (UARS) between 90 and 270 km seen in Lieberman et al. (2013). Equations 10 and 11 represent, respectively, the meridional and zonal wind components, given by:

$$U_x(z) = U_{x0}(z).cos\left(\frac{2\pi}{\lambda_x}(z - z_0) + \frac{2\pi}{T}\big(t - t_{x0}(z)\big)\right), \tag{10}$$

$$U_y(z) = -U_{y0}(z).sin\left(\frac{2\pi}{\lambda_y}(z - z_0) + \frac{2\pi}{T}\big(t - t_{y0}(z)\big)\right), \tag{11}$$

where $U_{x0}(z)$ and $U_{y0}(z)$ are the wind amplitudes at height $z$; $\lambda_x$ and $\lambda_y$ are the wavelengths for the respective meridional and zonal components of the diurnal, semidiurnal and terdiurnal tides; $z_0$ is a reference height (100 km); $t_{x0}(z)$ and $t_{y0}(z)$ are the wave phases; *T* is the tidal period, which can be diurnal (24h), semidiurnal (12h), or terdiurnal (8h).

It is important to mention that this work is the first study in which the terdiurnal tide was implemented in the MIRE model by using the SkiYMET wind profiles obtained from data collected at São João do Cariri. These data were used as representative of the winds over the observatory of Palmas since no other measurements at any closer station were available simultaneously with the ionosonde observations. Thus, the inferred neutral winds and atmospheric tides were included in Eqs. (10-11) of MIRE to simulate the *Es* layer dynamics over Palmas. A more detailed description about the radar system and the methodology used to compute the neutral wind parameters is available in Resende et al. (2017a).

### 3 Results and Discussion

### 3.1 Terdiurnal Tide Periodicities in *Es* Layer Occurrences

Figure 1 (panel a) shows the temporal variation (in UT) of the percentage occurrence of the *'h'*, *'c'*, and *'f/l'* types of *Es* layers during the summer solstice months of 2008/2009. The occurrences of the *'f'* and *'l'* types were grouped into one since both have nearly the same ionogram profile; however, the *'l'* type manifests only during the daytime, and the *'f'* type is typically nocturnal (Conceição-Santos et al., 2020). From December 2008 to February 2009 (summer), a total of 48 days of data was recorded by the ionosonde of Palmas, which corresponds to 2,859 ionograms. The daytime occurrence of *Es* layers was 65.58% (156.25 hours), while 34.42% (82 hours) occurred during nighttime. The $Es_{f/l}$ types were the most frequent, with an overall rate of 72.61% (173 hours).

Notice that a slight increase of the *Es* layer occurrence is observed between about 01-02 UT (LT = UT – 3.0h) with ~17% rate, and the $Es_{f/l}$ layers are predominant during nighttime. Then a drop occurs near dawn at 08 UT reaching occurrence rates of

less than ~7%. Approximately 8 hours after the first increase, the *Es* layers percentage of occurrence enhances markedly, reaching ~50% at 11 UT. This peak at 11 UT coincides with the increase in the occurrence rates of the '*c*' and '*h*' types of *Es* layers. The $Es_c$ and $Es_h$ layers are formed starting at dawn, after 10 UT, and cease at dusk around 20 UT. The $Es_c$ type corresponds to 13.47% (total of 32.08 hours), while the $Es_h$ type corresponds to a rate of 13.92% (total of 33.17 hours). It is clearly noticed that the $Es_h$ layer predominates over the other *Es* types between around 11-12 UT (~25% and ~23%, respectively). After this peak in the occurrence rate, *Es* types of the percentage of occurrence starts to decrease again and reaches values of ~7% between about 15-16 UT. However, after 17 UT, the occurrence rate starts to increase again and attains values of ~41% between around 19-21 UT. The three increases observed in the plot of occurrence rates of *Es* layer types with 8-h periodicities suggest a modulation associated with the terdiurnal tide during the summer period.

The analysis of the temporal occurrence of *Es* layer types over Palmas was extended for the autumn, winter, and spring seasons, as seen in Figure 1. In this figure, we show the occurrence rates for autumn (panel b), winter (panel c), and spring (panel d). As noted earlier for the summer, the $Es_{f/l}$ types were also predominant during nighttime, and the $Es_c$ and $Es_h$ types occurred only during daytime. At 12 UT in winter and between 11-15 UT in spring, the $Es_{f/l}$ layers were not predominant. In the autumn months, we observed three well-defined maxima in the occurrence rate of *Es* layer types, whereas in the winter (~9% at 02 UT) and spring (~5% at 02 UT) months, the first peak is not well defined. However, we observed clear second and third ones at around 10-12 UT and 19-20 UT, respectively, indicating the possible influence of the terdiurnal tide. The slight increase in the occurrence rate between around 03-04 UT during the spring season might suggest that besides the dominant terdiurnal tidal periodicities, there was also a weaker quarterdiurnal (6-h) oscillation affecting the *Es* layer development. By comparing the results shown in Figure 1, it is clearly noticed that another peak of ~47% was observed between around 10-11 UT during the spring season. Furthermore, the larger amplitudes of the *Es* layer occurrence rates associated with the terdiurnal tide modulation were observed during the summer and spring months.

The *Es* layers in Figure 1 ($Es_{f/l}$, $Es_c$, and $Es_h$) are formed by the wind shear mechanism due to the tidal wind component in the Southern Hemisphere (Arras et al., 2009; Haldoupis, 2012; Pancheva et al., 2013; Fytterer et al., 2014; Yu et al., 2019). The $Es_h$ type forms and disappears very fast compared to the $Es_{f/l}$ and $Es_c$ types, which indicates that this type of $Es_h$ layer may be formed mostly by molecular ions that have shorter lifetimes than metal ions (Plane, 2003; Plane et al., 2015). Also, it is possible to observe in Figure 1 that the $Es_h$ type shows a higher percentage in the peak rates between 11 and 12 UT, with a higher percentage occurring in summer. Oikonomou et al. (2014) explained that the meridional wind component is dominant over the zonal wind component above ~130 km in the formation of high *Es* layers. On the other hand, Conceição-Santos et al. (2020) analyzed the $Es_h$ type, and they concluded that the predominant wind direction controlling their formation and dynamics had equal importance for both the zonal and meridional components. Generally, the $Es_h$ layer performed a downward movement due to the wind dynamics, reaching heights at around 100 km, and then becoming the $Es_c$ or $Es_{f/l}$ layers. Haldoupis et al. (2006) mentioned that vertical shear in the zonal and meridional winds work together to generate the

descent of the $Es_h$ layers. In fact, the $Es$ layer types shown in Figure 1 strongly suggest that the formation mechanism of the $Es$ layers is the wind shear, emphasizing the main mechanism of the $Es$ layer development at low latitudes.

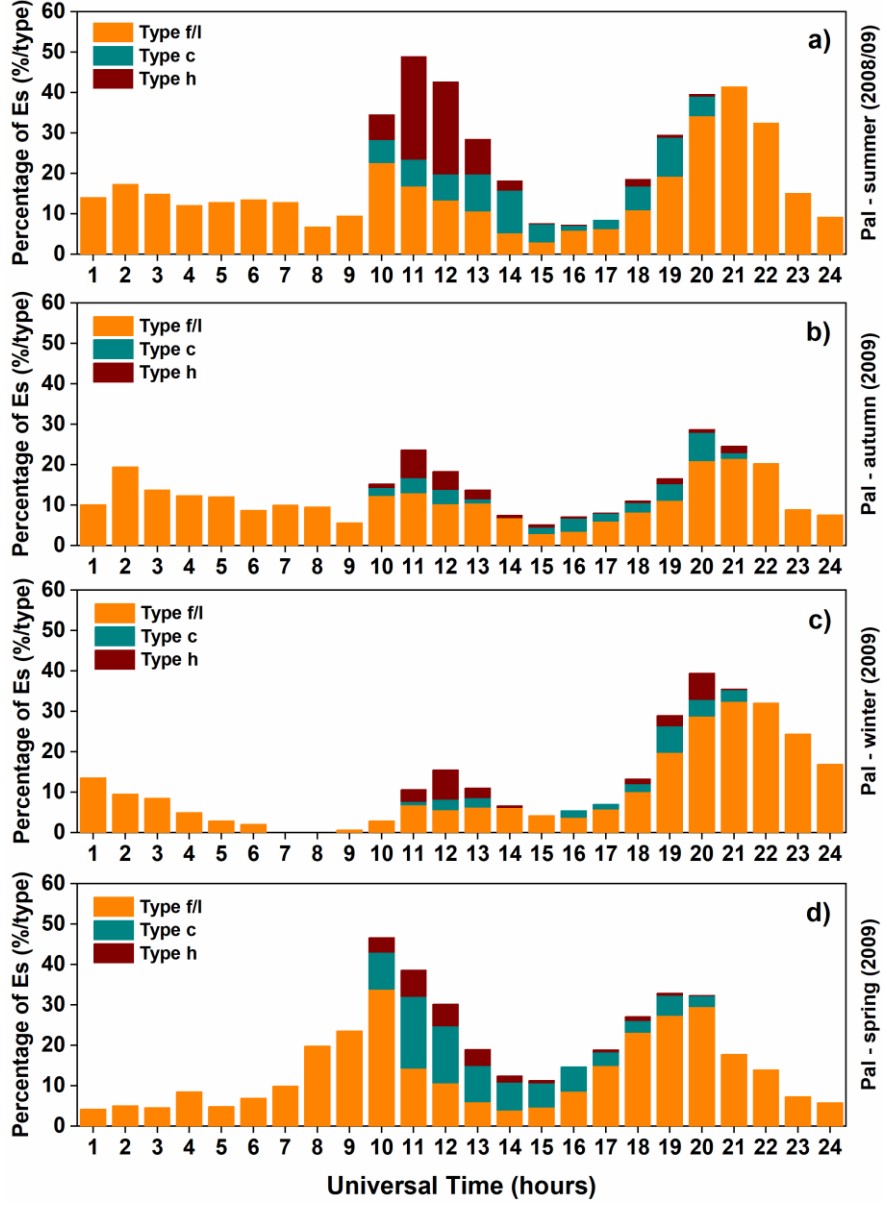

**Figure 1: Temporal variation of the *Es* layer occurrence rates divided in different types over Palmas during the summer (panel a) autumn (panel b), winter (panel c), and spring (panel d) months of the year 2008/09.**

Figure 2 depicts the temporal variation of the $Es$ layer ($h'Es$) for the '$h$', '$c$', and '$f/l$' types during the four seasonal periods at Palmas. It is seen from Figure 2 that the $Es_h$ layers were formed at higher altitudes between around 130 and 150 km, whereas the $Es_c$ layers were observed at altitudes ranging from 110 to 130 km. As expected, the $Es_c$ and $Es_{f/l}$ layers were mostly

observed at altitudes below 110 km, although during some specific hours at all seasons, they attained heights of ~130 km. The $Es_h$ layer tends to move down to lower heights in the E region, changing into types '*c*' or '*l*' when they reach 100 km

(Conceição-Santos et al., 2020). Another relevant aspect to be observed in Figure 2 is related to the distribution of *h'Es* during the times of maximum occurrence rate of the *Es* layer types, which is possibly associated with the terdiurnal tide. During the peaks in the frequency of occurrence of *Es* layers at around 10-12 UT and 18-20 UT, Figure 2 reveals for all seasons that the *Es* layers attained their highest altitudes. Otherwise, during the time of the first increment at around 01-02-UT, it was not evidenced significant changes in the heights of the $Es_{f/l}$ layers. However, it seems that the terdiurnal tidal oscillations had

some role in the formation of the $Es_h$ layers during the daytime.

As it is already known, tides are the major sources of vertical wind shear (Arras et al., 2009; Haldoupis, 2012; Pancheva et al., 2013; Fytterer et al., 2014; Yu et al., 2019; Jacobi and Arras, 2019). Thus, tidal-like structures are expected to influence the occurrence rates of *Es* layers strongly. As noticed in Figure 2, most of the *Es* layers are concentrated at heights below 120 km. According to Fytterer et al. (2014), the influence of the terdiurnal oscillations on *Es* formation and evolution is restricted to

heights between around 100-120 km. Thus, in this height range, the terdiurnal tidal oscillations may have a similar amplitude to the diurnal and semidiurnal tides (Zhao et al., 2005; Venkateswara Rao et al., 2011; Moudden and Forbes, 2013; Fytterer et al., 2014), and can also play an important role in modulating the *Es* layers. Bergsson and Syndergaard (2022) used the parameter *h'Es* to investigate the relationship of the *Es* layers at mid- and low-latitude with solar activity. They showed that *h'Es* vary with the seasons and significantly depend on solar activity in the nighttime period. Haldoupis et al. (2006) applied

the height-time-intensity (HTI) method to analyze the descent of the *Es* layers. The authors observed the formation of an *Es* layer at ~120-130 km that descends to ~100 km at dusk. Also, they observed that the formation of *Es* layers at heights above 125 km during nighttime has a higher descent rate than the daytime ones. Haldoupis (2011) and Andoh et al. (2020) mentioned that the zonal and meridional wind is equally efficient at ~125 km altitude where $v_i \sim \omega_i$. On the other side, at lower (higher) altitudes, the zonal (meridional) wind component becomes predominant in ion convergence by wind shear. This behavior

occurs because that at higher heights, the $v_i \ll \omega_i$, making the ions more magnetized, decreasing the efficiency of the zonal component, and increasing the contribution of the meridional component. Thus, the $Es_h$ layers in Figure 2 may indicate a predominance of meridional wind action, because these layers generally form at altitudes above 125 km and disappear quickly. The *Es* layer heights in summer shown in Figure 2 are in good agreement with the analyses by Yu et al. (2019) for both hemispheres using RO, where they observed a seasonal variation in the altitude-latitude distribution of the *Es* layer intensity

with the highest values at 125 km in the Southern Hemisphere summer and a wider latitudinal extent between 10° and 75° S. Other works, such as Haldoupis and Pancheva (2006), used the *h'Es* parameter to analyze the terdiurnal tidal oscillations over the *Es* layers. They concluded that the 8-hour oscillation is more pronounced in the *h'Es* than in the frequencies (*foEs*) and contributes to decreasing the *Es* layers with time.

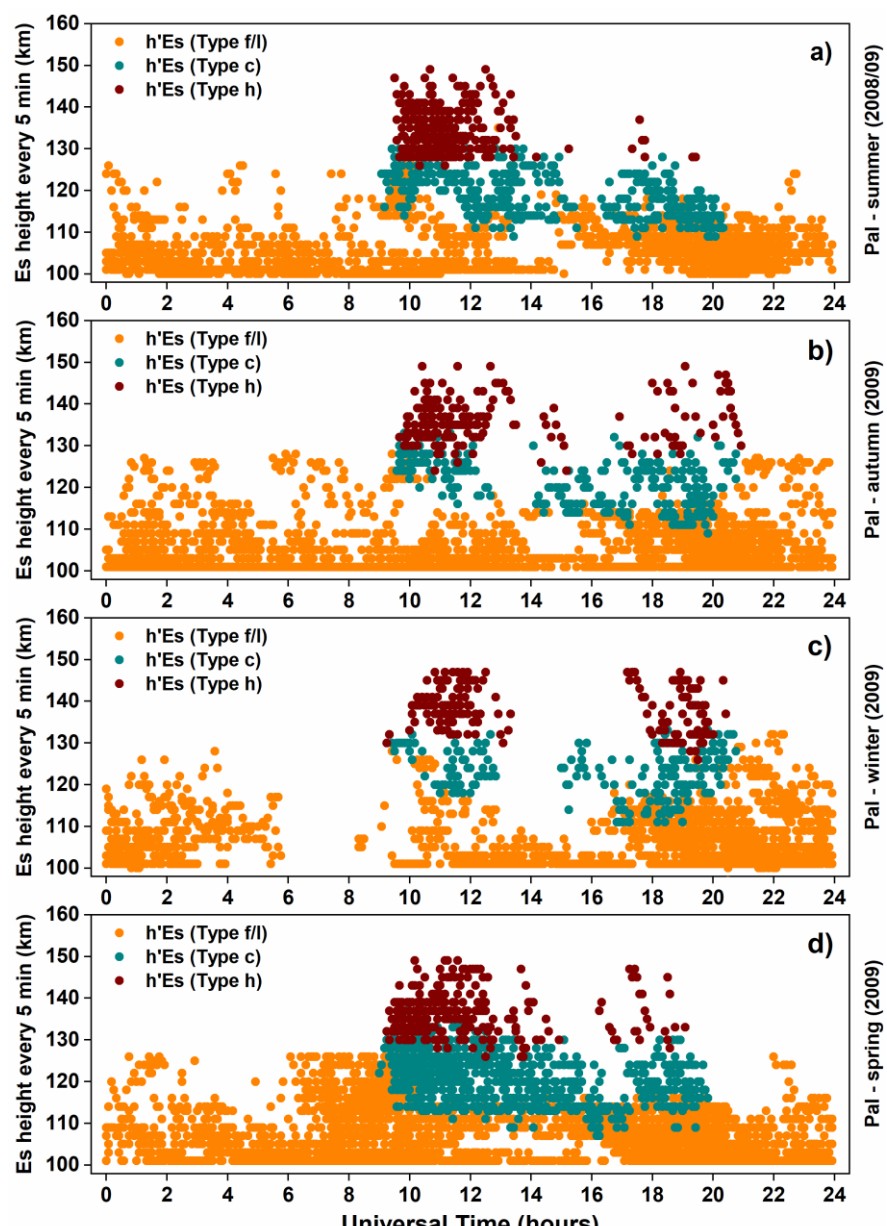

**Figure 2: Virtual height distribution of the *Es* layer types as function of time (in UT) over Palmas during summer (panel a), autumn (panel b), winter (panel c), and spring (panel d).**

All types of *Es* layers were grouped to calculate the temporal variation of the total percentage (black line) of occurrence of the

*Es* layers observed in Palmas during the four seasons. These results of total occurrence are shown in Figure 3. The three maximum values in the total occurrence rate of the *Es* layers during the summer and autumn months agree with the results observed before in Figure 1 (panels a and b). As already seen, these peaks are approximately 8 hours apart from each other, revealing a possible modulation by the terdiurnal tide. Analogously to what was observed in Figure 1 (panel c), during winter,

only two maxima in the total percentage of occurrence are noticed, being one at around 11-12 UT, and another at about 19-20 UT. As for the spring, a maximum in the occurrence rate was observed during the morning between approximately 09-10 UT, and ~8 hours later, another maximum was observed at dusk between around 18-20 UT. However, a small increase of ~8.5% was observed in the total percentage rate between around 03-04 UT during the spring, which reveals again the possible effect of a weaker 6-hour tidal mode embedded in the temporal occurrence rate of the *Es* layers.

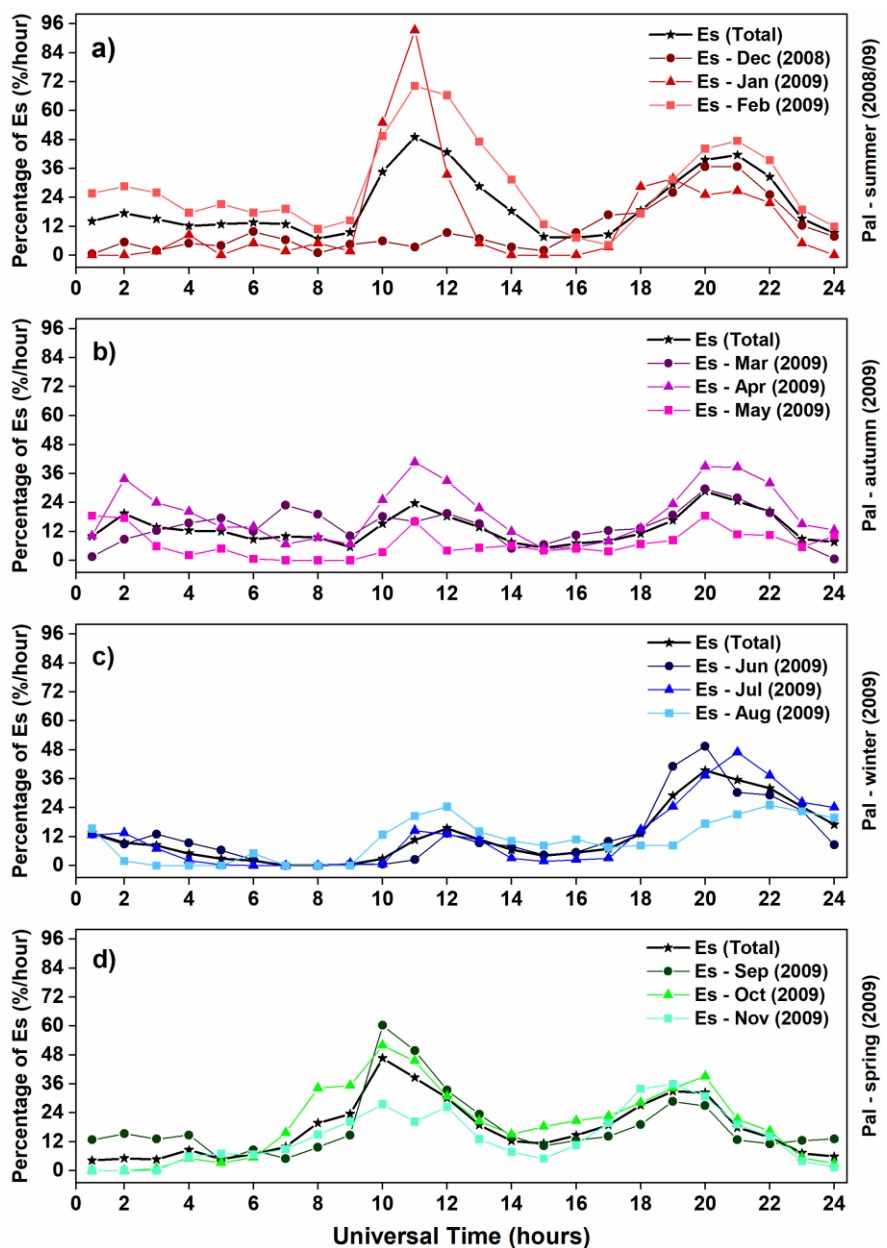

**Figure 3: Temporal variation of the *Es* layer occurrence over Palmas showed into individual month during the summer (panel a), autumn (panel b), winter (panel c), and spring (panel d).**

Figure 3 shows the monthly percentage of occurrences of *Es* layers divided by seasons. In summer (Fig. 3(a)), the terdiurnal modulation is dominant during February/2009. On the other hand, in December/2008 and January/2009, a first increase in the occurrence rate observed previously at around 01-02 UT is practically absent. This is probably related to the tendency for the amplitude of the migrating terdiurnal tide with zonal wavenumber 3 (TW3) to increase, generally from January to March within ±10° of latitude (Moudden and Forbes, 2013; Pancheva et al., 2013). Thus, the maximum peaks in the occurrence rates observed in February of 2009 are higher than those observed in December/January, owing to the fact that this month is when the summer solstice starts in the Southern Hemisphere. This result also agrees with Guharay et al. (2013) that it is shown a smaller terdiurnal tidal range for the December/January months and an increase of this component in February at low latitude stations over the Brazilian sector.

During autumn (Fig. 3(b)), the 8-h tidal oscillations are noticed in the results of April and May. Notice that the three peaks in the occurrence rates in April are higher than in the other months. Unlike the results shown in Figure 1, Figure 3(c) reveals a weaker increase in the percentage of occurrence (~12%) between around 01-03 UT, a clear second peak after 8-h, and a strong peak at approximately 19-21 UT. The total percentage of occurrence (Fig. (3)/black line) between 01-05 UT during winter was strongly influenced by the occurrence rate observed in August. Guharay et al. (2013) also found a significant increase in tidal amplitude in the autumn and minimum values for winter and early summer. Finally, in spring, the modulations associated with the terdiurnal tide are marked during September, October, and November.

The temporal variation of the total percentage of occurrence of *Es* layers was plotted in intervals of 20 min, as shown in Figure 4. The results for summer (red), autumn (purple), winter (blue), and spring (green) are shown as line plots. Then a nonlinear polynomial regression was applied to fit the occurrence rate curves for each season (black lines). The shadowed bands in the fitted curves denote the 95% confidence interval of the best-fitted parameters. It is possible to note that the curves of percentage rates are mostly within the 95% confidence range of the nonlinear curves. However, when the peaks in the occurrence rates are above the 95% level of confidence reveals the most significant effect of the terdiurnal tide on the formation and dynamics of *Es* layers.

It is worth noting from Figure 4 the 8-h periodicities in the fitted curves of all the seasonal periods. During winter, the first increase in the occurrence percentage at around 02-03 UT is not as pronounced as in the other seasons, but now it is more perceivable than Figures 1 and 2 and, also depicts the pattern seen in Figure 3 for June and July. According to Du and Ward (2010), the maximum amplitudes of the terdiurnal tide are confined between ±50° of latitude and tend to occur in winter between 80 and 100 km of altitude. So as the *Es* layers over Palmas were observed above 100 km, this may partly explain why the terdiurnal oscillations were less pronounced in winter.

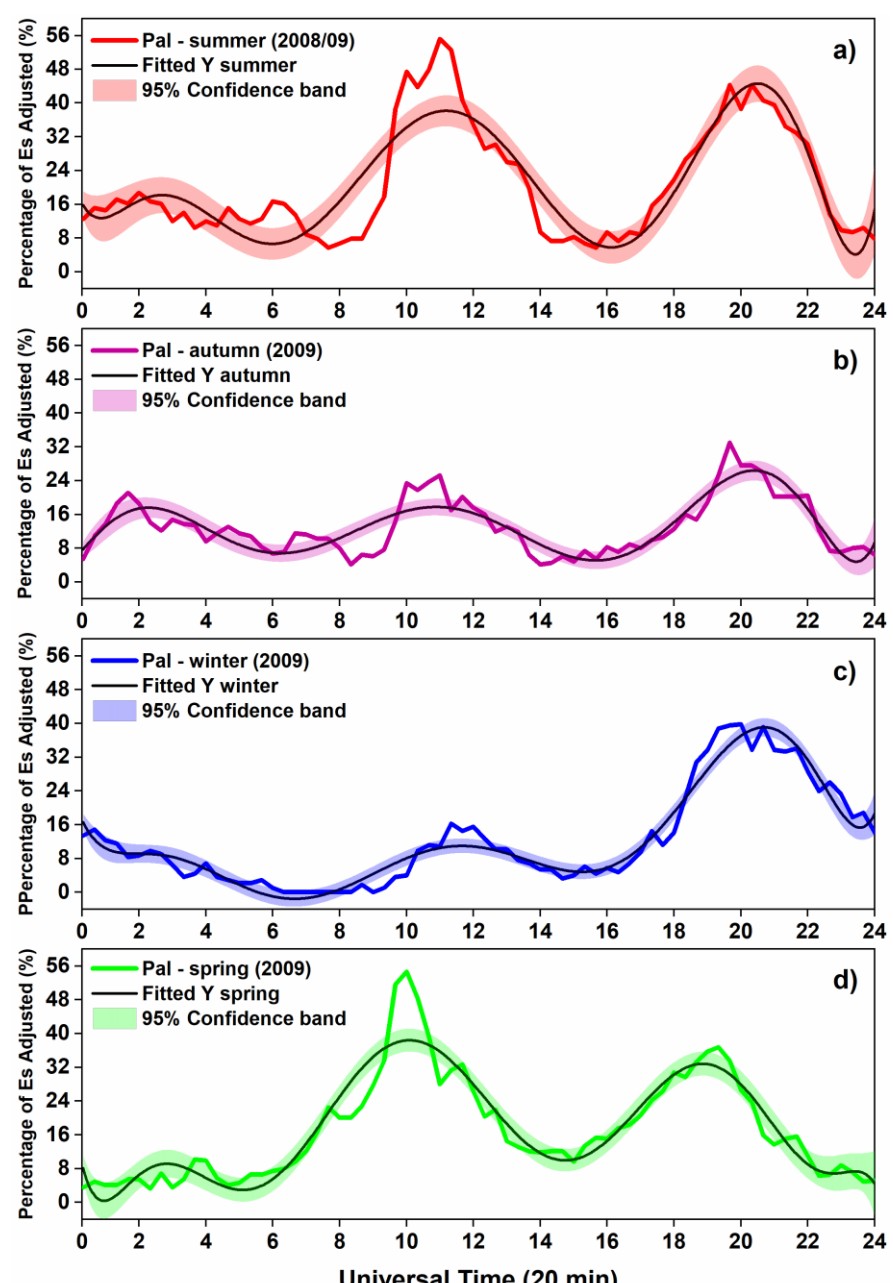

**Figure 4: Temporal variation of *Es* layers occurrences obtained in intervals of 20 min during summer (red line/panel a), autumn (purple line/panel b), winter (blue line/panel c) and spring (green lone/panel d) seasons. The black lines in the panels are the fitted curves of the occurrence rates by applying a nonlinear regression. The shadowed bands in the fitted curves denote the 95% confidence interval of the best-fitted parameters.**

The results in Figure 4 also agree with the study of Moudden and Forbes (2013), who analyzed 10 years of data and highlighted that migrating terdiurnal tide reaches higher amplitudes near the equator at altitudes above 100 km, but with higher amplitudes for the Southern Hemisphere. Pancheva et al. (2013) also analyzed 8 years (2002-2009) of terdiurnal tidal (TW3) oscillations

and found that between latitudes of ±10°, the amplitude of the terdiurnal tide at an altitude of ~90 km had a maximum in the month of February and that at 110 km altitude at low latitudes (±30°) had a maximum amplitude during the summer and winter. The 3D colormap surfaces in Figure 5 present the top frequencies of layer *Es* (*ftEs*) distributed per day (in UT) for the summer,

autumn, winter, and spring seasons. The color assigned in the graphs is related to the *Es* layer intensity. It is important to mention here that there were problems with the ionosonde equipment during the summer and the winter at the Palmas. Therefore, there are some gaps in the data with values corresponding to zero on the scale of values of the *ftEs* but not causing interference in the previous results presented here.

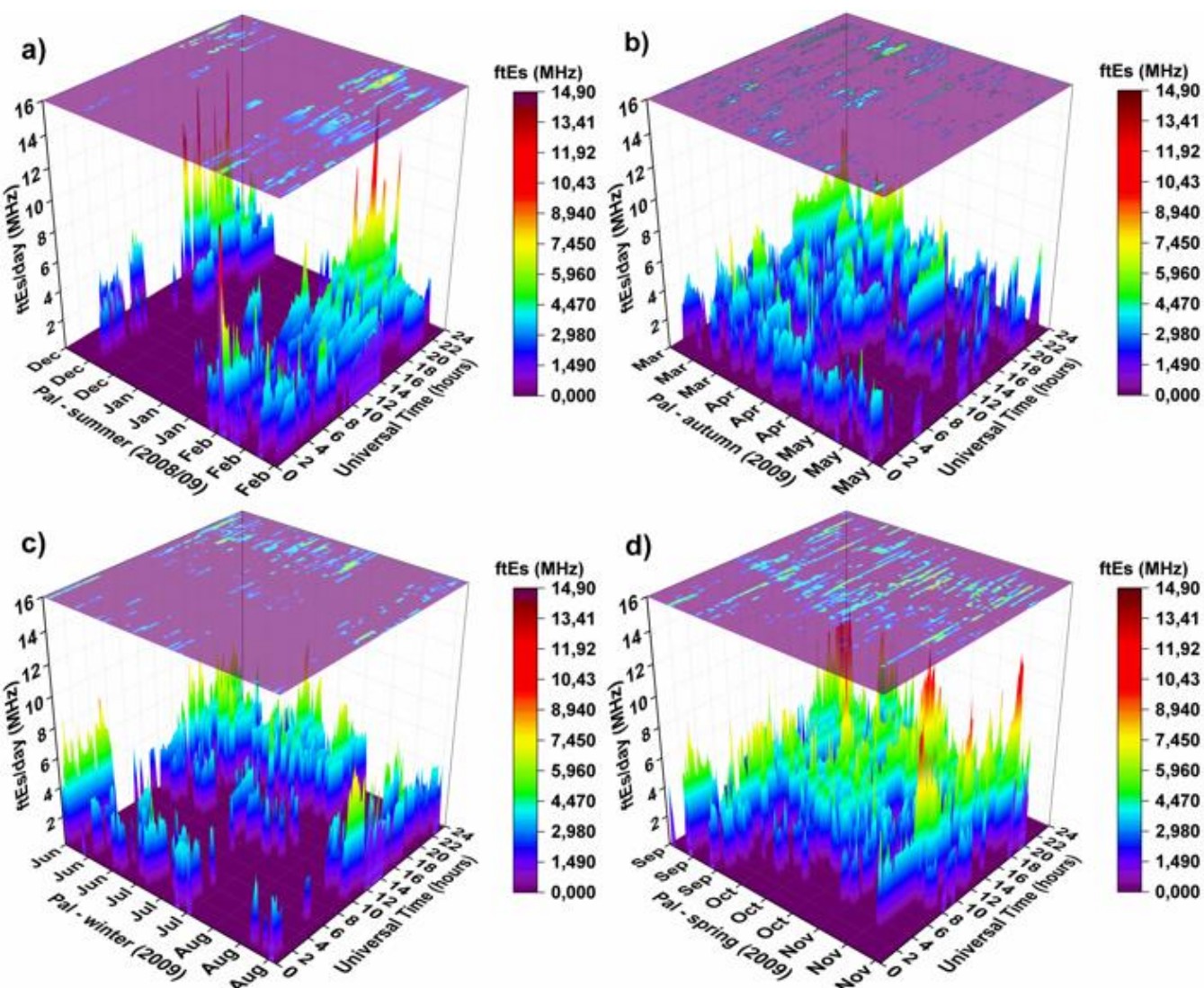

**Figure 5: The 3D colormap surfaces of the *Es* layer top frequency (*ftEs*) distributed per day (in UT) for the summer (panel a), autumn (panel b), winter (panel c), and spring (panel d) over Palmas.**

Figure 5 reveals that in autumn and winter, the *ftEs* magnitudes were lower than those observed during the summer or spring months. Such *ftEs* values are almost always below 10 MHz. On the other hand, it was observed that *ftEs* values were above 10 MHz during spring. Summer presented the highest *ftEs* values in the late afternoon, whose maximum intensity was 14.9 MHz (20 UT). This maximum intensity of the *ftEs* was detected in early February (02/06/2009) and some days in December 2008. We believe that the high values of the *ftEs* observed here in early February may be due to the event of sudden stratospheric warming (SSW) that occurred between late January and early February of 2009 (Wang et al., 2011; Fuller-Rowell et al., 2011). This SSW event had global effects on the semidiurnal (SW2), terdiurnal (TW3) tidal amplitudes, and planetary waves (PW1 and PW2) (Wang et al., 2011; Fuller-Rowell et al., 2011; Lin et al., 2012; Jin et al., 2012). It is well known that during a SSW the planetary waves can propagate vertically into the lower thermosphere and interact with the tides that dominate the dynamics in the ionospheric region. Thus, they can influence the development of the *Es* layers formed at this altitude (Liu and Roble, 2002; Lin et al., 2012; Fytterer et al., 2014). Wang et al. (2011) showed an SW2 tidal growth peaking after the SSW maximum and a TW3 decrease in the Southern Hemisphere. Subsequently, between 100-120 km altitude at 20-60º S, the rapid growth of tidal amplitude TW3 and a considerable decrease of SW2 occurred. The authors attributed to a transfer of energy from the SW2 to the TW3 tide in both hemispheres. Jin et al. (2012) showed that a notable increase in TW3 amplitude occurred at 115 km altitude after the SSW at low latitudes between 0-30° S. Therefore, the prominent peak in the *ftEs* (Fig. 5(a)) may indicate further evidence of the influence of the terdiurnal tide on the development of the *Es* layer. The results in Figure 5 are in good agreement with Yu et al. (2022) analysis. These authors compare the observational data using RO with a modeling the *Es* layers concerning the seasons. The authors show that summer and spring have more intense *Es* layer occurrences than autumn and winter in the Northern and Southern Hemispheres. Additionally, the results in Figure 5 are consistent with Tang et al. (2022), showing that the TW3 modulates the *Es* layers between 100-110 km of latitudes ±10°.

A power spectrum analysis with periodograms was performed on using the *fbEs* data (Fig. 6) to demonstrate that the periodicities present in the observational data are associated with tidal effects. The peak of the diurnal tide has been suppressed in Figure 6 for better visualization of the terdiurnal tide. The 6-h peak associated with a quarterdiurnal tide can be observed during the autumn, winter, and spring seasons.

Xu et al. (2014) showed that the nonlinear interaction between the planetary wave and the quarterdiurnal tide might be the source of the formation of the terdiurnal tides, and the nonlinear interaction between the diurnal tide and the terdiurnal tide may be the primary source of the quarterdiurnal tide. This may account for the increase/decrease in the 8 and 6-hour peaks observed in the periodogram analyses of the seasons in Figure 6. Guharay et al. (2013) also noted the presence of the quarterdiurnal tide at the latitude of São João do Cariri, which is a region of the Brazilian sector near Palmas, but indicated an adequate analysis would be needed to confirm this 6-hour oscillation. Lima et al. (2012) showed that the near 2-day planetary wave (QTDW) and other periodicities are also present in São João do Cariri region. Thus, it is possible that the occurrence of the 8 and 6 h oscillation shown in Figure 6 occurred by the nonlinear interaction between the planetary waves and the terdiurnal and quarterdiurnal tides, in addition to the nonlinear interactions between the diurnal and semidiurnal tides (Manson et al.,

1982; Teitelbaum and Vial, 1991; Pancheva, 2000; Chang et al., 2011; Chang et al., 2013; Gu et al., 2017). This topic is beyond the scope of this work due to the lack of wind data, but it can be explored in the future.

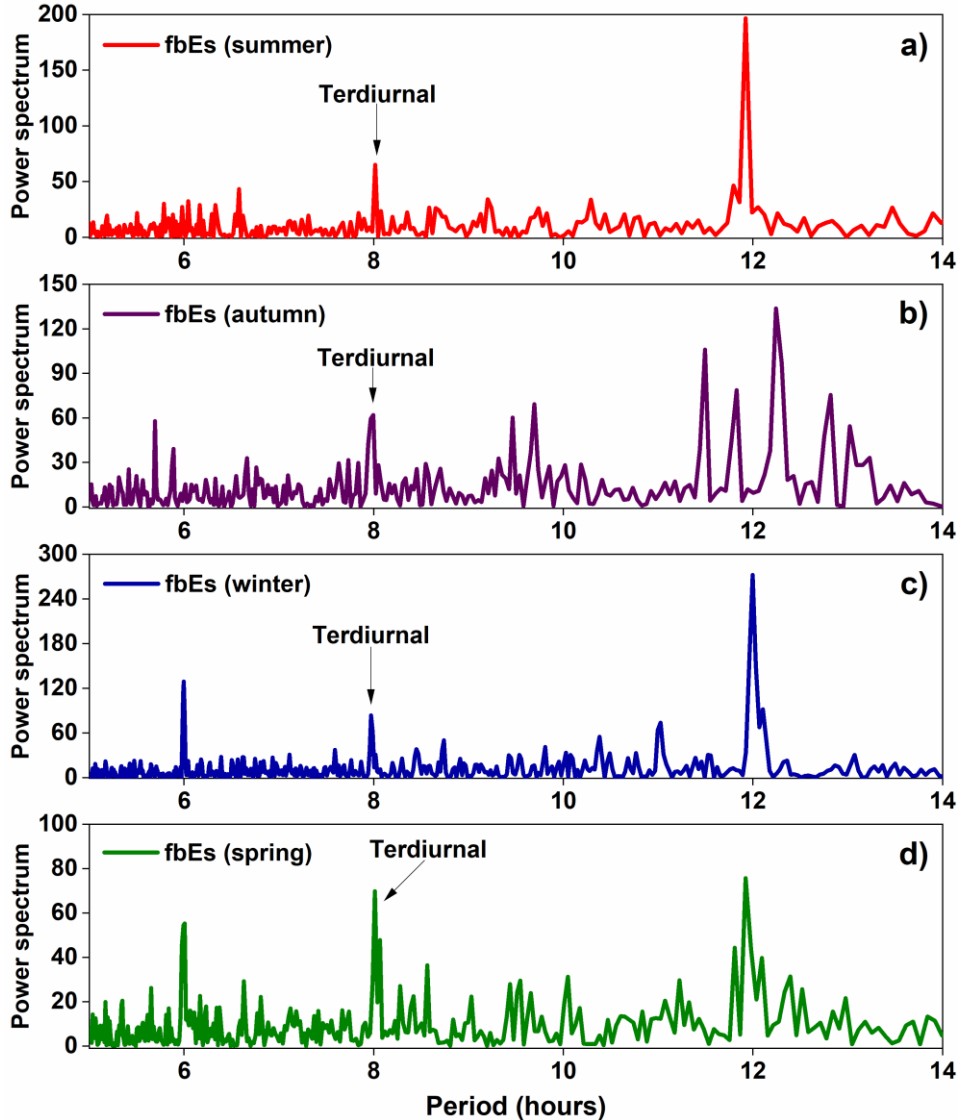

**Figure 6: Periodogram analysis of the *fbEs* with the power spectrum representations of a) summer, b) autumn, c) winter, and d) spring. The best-fitting peaks correspond to the 99% confidence interval of the *fbEs* parameters.**

Lastly, the results of the *Es* layer intensity do not provide a conclusion that terdiurnal tide is influencing the densities, as is 345 shown in height. However, we found some correlation between the strong *Es* layer and the most terdiurnal modulation, as seen during the summer. Thus, we used a model to analyze the terdiurnal tide role in the *Es* layer behavior that will be present in the following section.

## 3.2 E Region Simulations

The simulations of the ionospheric electron density profiles (in logarithmic scale) between 86 and 140 km concerning the Universal Times are shown in Figure 7. We considered in the simulations the three tidal structures (semidiurnal, diurnal, and

terdiurnal periodicities) for the zonal and meridional wind components. The panels on the left column of Figure 7 refer to the results of diurnal and semidiurnal tides (D+S), while the panels on the right column show the simulation results considering the diurnal, semidiurnal, and terdiurnal tides (D+S+T). We selected data from different months to represent each seasonal period: December 2008 (summer/panels a), April 2009 (autumn/panels b), July 2009 (winter/panels c), and October 2009 (spring/panels d). These months were chosen because, during the period analyzed here, they presented better diurnal,

semidiurnal, and terdiurnal wind data estimations from meteor radar observations at São João do Cariri, and there is already a study on the terdiurnal tide with data from this station in the Brazilian sector that can be found in Guharay et al. (2013). As we mentioned before, these wind data were used as input to the MIRE model. The thin traces of enhanced electron density seen in the plots of density profiles in Figure 7 denote the presence of the *Es* layers.

In the plots of electron density profile shown in Figure 7(a) for December (summer), it is possible to see that when only diurnal

and semidiurnal tides (D+S) are considered in the simulations, the *Es* layer is formed at 16 UT just below 130 km altitude. This *Es* layer tends to move downward throughout the night and reaches an altitude of ~115 km near dawn at 09 UT. During the day, it continues to descend slowly until reaching ~110 km at around 16 UT. The density of this *Es* layer has a maximum value of ~$10^{5.1}$ electrons/cm$^3$. When adding the terdiurnal tide (D+S+T) in the simulations, the results in the right panel of Figure 7(a) clearly reveal that the layer is formed at higher altitudes above 135 km at around 14 UT. Analogously to what was

observed from the (D+S) simulation, the *Es* layer descended continuously and reached ~110 km 24-hours later. It is also noted an intensification of the maximum *Es* layer density to ~$10^{5.8}$ electrons/cm$^3$.

The results of the MIRE simulations for April (autumn) are presented in Figure 7(b). The D+S simulation shows a thick *Es* layer starting at around heights of more than 138 km and descending continuously, reaching ~120 km altitude at 24 UT. The maximum density of this *Es* layer is about ~$10^{5.0}$ electrons/cm$^3$. By adding the terdiurnal tide (D+S+T) into the simulation, the

results show that the most intense *Es* layer forms at about ~140 km altitude at 03 UT and moves downward. Notice that along almost its entire descent trajectory, the layer density is notably larger (~$10^{5.2}$ electrons/cm$^3$) compared to the D+S simulation. The panels in Figure 7(c) for July (winter) show similar results from the D+S and the D+S+T simulations regarding the height of formation and descent of the *Es* layers. In both simulations, the *Es* layer is formed at around 06 UT at heights of ~140 km. As the layer descends throughout the day, the electron density increases ~$10^{4.5}$ electrons/cm$^3$ for the D+S simulation and ~$10^{5.0}$

electrons/cm$^3$ for the case D+S+T simulation. From 22 UT until around 04 UT, the layer stops descending, becomes thicker, and its electron density decreases to ~$10^{4.0}$ electrons/cm$^3$. Furthermore, it presents a slight rise from ~117 to 122 km. Then the *Es* layer starts to descend again until pre-sunrise hours.

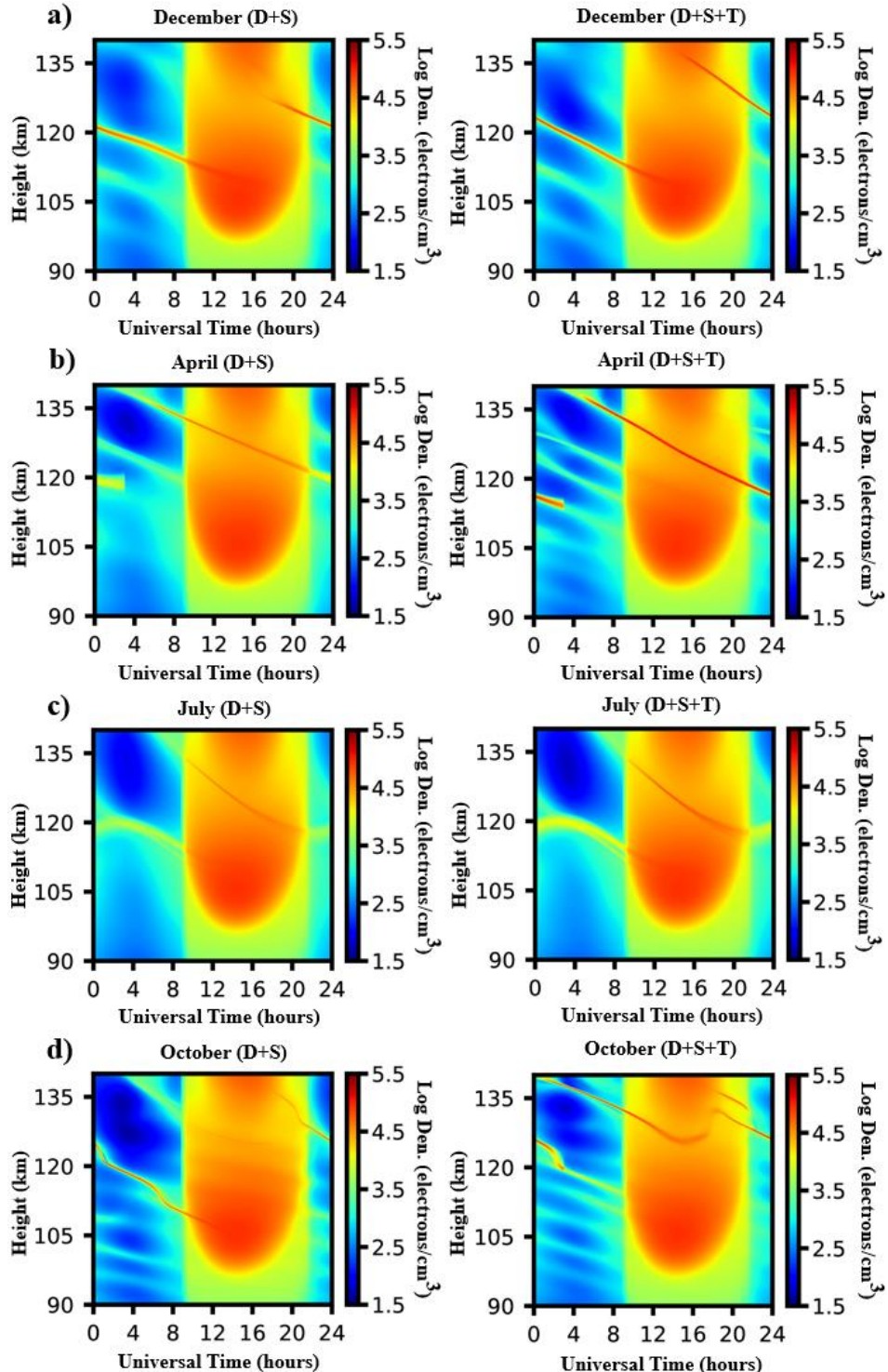

**Figure 7: Density profiles of the *Es* layers simulated by MIRE during (a) December (summer), (b) April (autumn), (c) July (winter), and (d) October (spring).**

Finally, the MIRE simulations for the month of October (spring) present in the panels of Figure 7(d) a very distinct diurnal behavior between the D+S and D+S+T simulations. The *Es* layer in simulation due to D+S is formed during daytime at ~140 km. This layer moves to downward, reaching ~125 km at 24 UT. The results show that this layer continues to move downwards throughout the night and remains until around 12 UT when it attains ~107 km. A marked feature observed along the descent course of this layer is the presence of oscillations in its height. The maximum electron density was ~$10^{4.9}$ electrons/cm$^3$ and reached ~$10^{5.3}$ electrons/cm$^3$ with the D+S+T component. Such oscillations seen in the traces of *Es* layers for both plots of panels 7(d) have already been reported by Conceição-Santos et al. (2020) as possible interactions between gravity/planetary waves and tides.

Additionally, according to Lilienthal et al. (2018), the terdiurnal tide can only occur if both the diurnal and the semidiurnal tides show considerable amplitudes. This is consistent with our results since the terdiurnal tide alone was not able to generate *Es* layers in the simulations (not shown here). Moudden and Forbes (2013) compared diurnal, semidiurnal, and terdiurnal tides, and they stated that there is a strong possibility that the non-linear interaction between diurnal and semidiurnal tides is the main source of terdiurnal tide formation. Thus, depending on altitude and latitude, the terdiurnal tide can exhibit amplitudes with significant magnitude, and this can play an important role in the dynamics and modulations of *Es* layer densities (Moudden and Forbes, 2013; Pancheva et al., 2013; Fytterer et al., 2013; 2014).

By using radio occultation (RO) data, Fytterer et al. (2014) performed a global analysis of the effect of the terdiurnal tide on the occurrences of the *Es* layers. They used data collected in the period from 2006 to 2012 during the four seasonal periods for latitudes of ±60° and altitudes ranging between 85 and 115 km. The authors found two amplitude maxima in the terdiurnal tide signature over the *Es* layers during the solstice that maximize between 10° and 40° latitude in both hemispheres. In addition, they also found similar behavior for low-latitude equinox conditions in both hemispheres for altitudes above 100 km. Thus, the intensification of the *Es* layer densities obtained here from MIRE simulations are in accordance with the observations in Fytterer et al. (2014). In summer and spring, the simulations showed a more intense *Es* layer variation when added the 8-h tide. This behavior can cause high *ftEs* to be observed in such seasons. Finally, our results show that the inclusion of the terdiurnal tide caused an increase in the electron density of the *Es* layers for all seasonal periods.

The simulations show that the electron density of the *Es* layer increases when we include the terdiurnal tide. To see this behavior better and to compare it with the ionosonde data, we show Table 2. This Table contains the results of maximum electron density considering D+S and D+S+T simulations, and the maximum average daily electron density observed from the ionosonde data for the same months used in the MIRE. Also, we included the maximum electron density of the most pronounced peak in our observational data. We used the *ftEs* parameter to calculate the electron density. The density is given in electrons/cm$^3$ on the logarithmic scale.

**Table 2:** Comparison of the simulated *Es* layer densities in MIRE with the D+S, D+S+T component and the maximum average daily density from the observed ionosonde data.

| Month | Electron Density Peak D+S (MIRE) | Electron Density Peak D+S+T (MIRE) | Electron Density Peak (Ionosonde and MIRE) | Electron Density Peak (Ionosonde – Most pronounced peak) |
|---|---|---|---|---|
| December | 5.09 | 5.81 | 5.77 (Dec) | 5.51 (Feb) |
| April | 4.93 | 5.22 | 5.43 (Apr) | 5.43 (Apr) |
| July | 4.50 | 4.92 | 5.33 (Jul) | 5.33 (Jul) |
| October | 4.92 | 5.3 | 5.66 (Oct) | 5.37 (Set) |

In general, in Table 2, it is possible to notice that the maximum electron density observed in ionosonde data agreed with the simulations considering the terdiurnal tide component. Notice that there are close values between the densities of the *Es* layer simulated in MIRE and the value of the maximum mean daily density observed in the ionosonde data in December. The value of the maximum average daily density observed in February, the month that best represented the 8-hour oscillation in Figure 3, is also in good agreement with the result simulated in MIRE. Also, the months of April and July coincided with the best 8-hour oscillation in the percentages of *Es* layers in Figure 3 and simulations for D+S+T. The densities of the *Es* layers modeled in MIRE with D+S+T are also in good agreement with the densities of the *Es* layers observed with the *ftEs* values in Figure 5, where higher values are shown for summer and spring and lower values for fall and winter. Finally, October followed the pattern found in December and April, showing good agreement between the simulated *Es* layer densities with the terdiurnal tidal component and the maximum mean daily density observed in the ionosonde data. Thus, we notice that the terdiurnal component is important to the *Es* layer formation around the months analyzed in this study.

**4 Conclusion**

This paper presented an analysis of the effect of terdiurnal tidal modulation on the occurrence rate of the *Es* layers observed during the years 2008/09 in Palmas, a low latitude station located in Brazil. We used ionosonde data and simulations to investigate the effect of terdiurnal (8-h) tidal periodicities on the formation of the *Es* layers during different seasonal periods. The main results of this study are summarized as the following:

1. The $Es_{f/l}$ is the most common type found in all seasons, and types $Es_c$ e $Es_h$ appear exclusively in the daytime over Palmas, agreeing with the previous results shown in Conceição-Santos et al. (2020). This behavior means that the meridional and zonal tidal wind component is acting the most time over this station.

2. We observed three peaks in the *Es* layer behavior for all seasons, revealing that the terdiurnal tide (8 hours) can modulate their formation. This fact is very clear during the summer and spring. In summer, the 8-h modulation is controlled mainly by the *Es* layer occurrence in February, probably due to the terdiurnal tide having lower amplitudes in the summer onset. Although

autumn shows an 8-h modulation in the *Es* layer occurrence, the first peak is not well-defined in March. During the winter, August has a poorly rate in the terdiurnal oscillation of the *Es* layer occurrence compared to June and July.

The 8-h modulation in the *Es* layer development over Palmas was confirmed using a nonlinear regression for each season. It is worth noting the 8-h periodicities in the fitted curves of the *Es* layer rates in all seasonal periods evidencing a possible effect of the terdiurnal tide on the development of *Es* layers. This behavior is less pronounced during the winter, agreeing with the maximum amplitudes of the terdiurnal tide occurring between 80 and 100 km of altitude in this season. Therefore, as the *Es* layer appeared above 100 km, we do not have an 8-h clearly on our data here.

3. The *ftEs* had lower values in the autumn and winter than those observed during the summer or spring months (below 10 MHz). Summer presented the highest *ftEs* values, that can be due to the Gradient Drift instability of the Equatorial Electrojet Current (EEJ), which is effective during the magnetic storms in the boundary equatorial magnetic sites, as is the case of the Palmas region. The disturbed electric fields can superimpose the wind shear mechanism during disturbed periods. As in this work, we do not separate the quiet and disturbed days, a more in-depth study of these *Es* layer development will be necessary, and we intend to present it in the future. The month of February showed a prominent peak at ~20 UT of 14.9 MHz that may be associated with the Sudden Stratospheric Warming (SSW) that occurred in early 2009 and had global influences on atmospheric phenomena.

4. The results of the periodogram analysis using the *fbEs* data showed the presence of the terdiurnal tide in all four seasons, indicating that this tidal component contributed to modulate the *Es* layer development over Palmas. Besides the expected diurnal and semidiurnal tidal components, we also found in this analysis the 6-hour tidal periodicity, which clearly appeared in the power spectrum signatures during autumn, winter, and spring seasons.

5. The results show that the densities of the *Es* layers simulated in MIRE are more intense when the terdiurnal tidal component is added to the diurnal and semidiurnal tides, agreeing with the observations previously reported in work of Fytterer et al. (2014). Comparisons between simulated and observed *Es* layer peak electron densities have shown an overall good agreement during the four months analyzed here.

**Acknowledgments**

PAF thanks Coordenação de Aperfeiçoamento de Pessoal de Nível Superior (CAPES) for student funding under grant number 88887.372542/2019-00. The authors thank the São Paulo Research Foundation (FAPESP) for the support received under grants 2019/19225-9, 2019/09361-2, 2016/22634-0, and for funding the São João do Cariri Meteor Radar which is operated in cooperation of National Institute for Space Research (INPE) and the Campina Grande Federal University (UFCG). We also thanks to Fundação de Pesquisa do Maranhão (FAPEMA), SECTI and the Government of the State of Maranhão, Brazil, for partial funding of this research under process number BD-02689/20. MTAHM is grateful for the financial support received by the Conselho Nacional de Desenvolvimento Científico e Tecnológico (CNPq) under process number 314261/2020-6. LCA Resende and V.F. Andrioli thanks the China-Brazil Joint Laboratory for Space Weather (CBJLSW), National Space Science

Center (NSSC), Chinese Academy of Sciences (CAS) for supporting their Postdoctoral fellowships. Finally, the authors thank Drs. Ricado Buriti and Washington Lima for their efforts to continue the operation of the instruments at São João do Cariri and Palmas, respectively.

## Data availability

The ionosonde data and the meteor radar data are available for download at UNIVAP's website in the link: www1.univap.br/ionosfera/Paper_Data/2022_Pedro_angeo.zip.

## Author contributions

PAF and all the coauthors designed the experiments and carried them out. PAF, MTAHM, LCAR and VFA analyzed and interpreted the data. LCAR and AJC developed the MIRE model and performed the simulations with contributions of PAF. PRF and VGP were responsible for ionosonde operation and its data management. PPB and VFA were responsible for meteor radar operation and its data processing. PAF and MTAHM prepared the paper with contributions of all coauthors.

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
