# Peer review of "Effects of the terdiurnal tide on the Sporadic-E layers (*Es*) development at low latitudes over the Brazilian sector"

_Annales Geophysicae, 2022_

## Author Comment (AC1)

**Responses to the Comment and/or Suggestions from Referee 1**

Comments on "Effects of the terdiurnal tide on the Sporadic-E layers (Es) development at low latitudes over the Brazilian sector" by Pedro Alves Fontes et al., Ann. Geophys. Discuss., https://doi.org/10.5194/angeo-2022-17-RC1, 2022.

Using the ionosonde and meteor radar observations and the MIRE model, this study investigated the effects of terdiurnal tide on the Es layers at low latitudes in the Southern Hemisphere. The paper suggested that the terdiurnal tide could help in the formation of the Es layers in the Brazilian sector. The results could improve our understanding of the formation of Es layers in the lower latitudes. However, they are major issues that need to be clarified first.

*Thank you very much for the revision given by the referee. We have carried out a revision of the manuscript considering all the referee's comments.*

**Comments:**

1. The production and loss terms used in the model should be given.

**Response:**
Ok. We include information and Table 1 with the equations and rate coefficients in this new version (See lines 124 to 129).

2. The paper mentioned that the wind data used in MIRE were obtained by the meteor radar. The altitude range for the meteor radar is from 80 to 100 km while the height range of interest in this study is between 86 and 140 km. The authors mentioned that they expanded the wind equation to 120 km. It is not clear to me how the authors obtain the neutral wind data from 100 to 140 km and the accuracy of the wind data in this altitude range. The neural wind data is very important in simulating the formation of the Es layers.

**Response:**
We apologize for not making this clear. In fact, the measurements available from the meteor radar provide data from 80 to 100 km only. However, the height range of interest

in MIRE to simulate the E region and the Es layer dynamics is from 86 to 140 km. Thus, to extend the wind equations up to 140 km, we used a fitting function for each parameter as proposed in Resende et al. (2017a; 2017b). In this work, the Lorentz curve (according to the equation below) was used in this study for wind amplitude, considering the theory about the wind behavior (Lindzen and Chapman, 1969; Forbes and Garret, 1979).

$$U_{(x0,y0)}^{(24,12,8)}(z) = \mu_{(x0,y0)} + \frac{2.A_{(x0,y0)}}{\pi} \cdot \frac{\varphi_{(x,y)}}{4.\left(z - h_{(x0,y0)}\right)^2 + \varphi_{(x,y)}^2},$$

where $\mu_{(x0,y0)}$, $A_{(x0,y0)}$, $\varphi_{(x,y)}$ and $h_{(x0,y0)}$ are the fitted parameters for the meridional and zonal components of the daytime (24 h), semidiurnal (12 h) and terdiurnal (8 h) tides. Additionally, the wind phases were obtained by a linear fitting of the observational data, for São João do Cariri region. And the vertical wavelengths can be used from the respective wave phase equations by multiplying the linear coefficient of the fitted curves by the corresponding tidal period 24 h for diurnal tide, 12 h for the semidiurnal tide, and 8 for the terdiurnal (Buriti et al., 2008). It is important to mention here that Conceição-Santos et al. (2019) proposed a modification in the $\lambda_{x,y}$ of the extrapolated winds, keeping them constant between 100 and 120 km, and tending to infinity between 120 and 140 km. A detailed description about the radar system and the methodology used to compute the neutral wind parameters is available in Resende et al. (2017a). Finally, we used the Es layer density profiles of the MIRE for the months of December (summer), April (autumn), July (winter) and October (spring) as representative of the seasons.

All these information was included in lines 143 to 156 in this new version.

3. I think that to study the seasonal variation, multiple years of data are needed. This paper only used one year of data. I suggest that the authors use a larger dataset.

**Response:**

We appreciate and understand what the authors pointed out. However, in this study, we believe that one year is enough to have an idea of the Es layer behavior, as seen in Resende et al. (2017a; 2017b) and Conceição-Santos et al. (2019). The tidal wind pattern does not have significant changes each year. The GSWM model, for example, gives us the monthly parameters representing all the years. Furthermore, the inclusion of more years of data

takes a long time to analyze since the interpretation and reduction of the ionosonde data are not carried out from automatic program. Finally, the purpose of this study is to show that in addition to the diurnal and semidiurnal tides, there is also an influence of the terdiurnal tide in the Es layer formation over the Brazilian station. We highly respect the reviewer's opinion, but we believe that our results are sufficient to analyze the development of the Es layers in this low latitude sector.

4. Figures 1 to 5 are not new to me. At least, the authors should make comparisons with previous studies.

**Response:**

We thank the reviewer for the suggestions. Regarding Figures 1 and 2, we have merged them into a single figure (classified as Figure 1 now). In the discussion about Figure 1, we added other references to improve text. Also, we discussed the importance of analyzing the Es layers concerning the types, showing the mechanism of Es layer formation at latitude. Additionally, we included a discussion of the Esc and Esh types (lines 201 to 213).

Also, with respect to Figure 2 (before Figure 3), we have added discussions showing the relationship of the h'Es with solar activity and the terdiurnal tide (lines 231 to 240).

Figures 4 and 5 have been merged into a single figure. It is now Figure 3. In this figure, we kept the discussion by highlighting the months that best represented the 8-hour oscillations in the sum of the Es layer types in each season.

5. Figure 8: the comparisons should be made among observations, simulation1 (D+S), and simulation2 (D+S+T).

**Response:**

Ok! At the end of the discussion of this figure (now Figure 6), we have added a table where we show a clear comparison of the densities of the Es layers simulated in MIRE with the D+S and D+S+T components and the maximum daily average density for those months found in the ionosonde data. In the last column of Table 2, we also included the maximum average daily densities for the months that had the most pronounced peak. Additionally, we discussed these results in this new version (lines 363 to 377).

6. Terdiurnal tides in the low latitudes have been widely studied. The literature research could be done better.

**Response:**

We thank the reviewer for the suggestion. The references about the terdiurnal tide have been improved in the introduction, with the addition of two paragraphs (lines 40 to 73) where we show the two theories of tidal generation currently discussed in the literature. We also added new discussions to the results and added some more references on the topic.

Finally, we would like to thank Referee 1 for the suggestions and corrections to improve the manuscript.

**References**

Buriti, R. A., Hocking, W. K., Batista, P. P., Medeiros, A. F., and Clemesha, B. R. Observations of equatorial mesospheric winds over Cariri (7.4° S) by a meteor radar and comparison with existing models, *Annales Geophysicae*, 26, 485–497, https://doi.org/10.5194/angeo-26-485-2008, 2008.

Conceição-Santos, F., Muella, M. T. A. H., Resende, L. C. A., Fagundes, P. R., Andrioli, V. F., Batista, P. P., Pillat, V. G., and Carrasco, A. J. Occurrence and Modeling Examination of Sporadic- E Layers in the Region of the South America (Atlantic) Magnetic Anomaly, *Journal of Geophysical Research: Space Physics*, 124, 9676–9694, https://doi.org/10.1029/2018JA026397, 2019.

Forbes, J. M., and Garrett, H. B. Theoretical studies of atmospheric tides. *Reviews of Geophysics*, 17, 1951-1981, https://doi.org/10.1029/RG017i008p01951, 1979.

Lindzen, R. S., and Chapman, S. Atmospheric tides, *Space Science Reviews*, 10, 1415–1422, https://doi.org/10.1007/BF00171584, 1969.

Resende, L. C. A., Batista, I. S., Denardini, C. M., Batista, P. P., Carrasco, A. J., Andrioli, V. F., and Moro, J. The influence of tidal winds in the formation of blanketing sporadic e-layer over equatorial Brazilian region, *Journal of Atmospheric and Solar-Terrestrial Physics*, 171, 64–71, https://doi.org/10.1016/j.jastp.2017.06.009, 2017a.

Resende, L. C. A., Batista, I. S., Denardini, C. M., Batista, P. P., Carrasco, A. J., Andrioli, V. de F., and Moro, J. Simulations of blanketing sporadic E-layer over the Brazilian sector

driven by tidal winds, *Journal of Atmospheric and Solar-Terrestrial Physics*, 154, 104–114, https://doi.org/10.1016/j.jastp.2016.12.012, 2017b.

---

## Author Comment (AC2)

**Responses to the Comment and/or Suggestions from Referee 2**

Referee comment on "Effects of the terdiurnal tide on the Sporadic-E layers (Es) development at low latitudes over the Brazilian sector" by Pedro Alves Fontes et al., Ann. Geophys. Discuss., https://doi.org/10.5194/angeo-2022-17-RC2, 2022.

The topic of this paper is the observation of sporadic E (Es) layers with a low-latitude ionosonde station in Brazil. The authors focus on the terdiurnal tidal component that they extract from Es occurrence and other related parameters provided by the ionosonde. The topic is interesting and important to the community since until today there is still a lack of understand the ion-neutral coupling processes and the exact contribution of the tidal species to Es formation. However, I feel the presentation must be improved before the paper can be published in Annales Geophysicae. Please find my detailed comments below:

*Thank you very much for the revision given by the referee. We have carried out a revision of the manuscript considering all the referee's comments.*

**Major points:**

1. It is interesting to see that different types of Es layers appear during different times of the day. However, from my point of view Fig. 1,2 and Fig.3 contain almost the same information. Only from figure 3 the reader can get a rough estimation on how frequent each Es type appears. Of course, it is also beneficial to present absolute numbers. I recommend to combine these Figures but keep Figure 3 and adjust the text accordingly.

**Response:**
We thank the reviewer for the suggestions. Regarding Figures 1 and 2, we have merged them into a single figure (classified as Figure 1 now). In the discussion about Figure 1, we added other references to improve text. Also, we discussed the importance of analyzing the Es layers concerning the types, showing the mechanism of Es layer formation at latitude. Additionally, we included a discussion of the Esc and Esh types (lines 201 to 213).

Also, with respect to Figure 2 (before Figure 3), we have added discussions showing the relationship of the h'Es with solar activity and the terdiurnal tide (lines 231 to 240).

2. Same applies for Figure 4 and Figure 5. The information both plots contain are redundant and I recommend to delete Figure 4 because all necessary information are presented already in Fig. 5. You may think of adding a 4th line to Fig 5 representing the seasonal mean from Figure 4.

**Response:**

Ok! Figures 4 and 5 have been merged into a single figure. It is now Figure 3. In this figure, we kept the discussion by highlighting the months that best represented the 8-hour oscillations in the sum of the Es layer types in each season.

3. Starting from line 126: You identified a 8-h structure in the Es data. But the rates during the night are very low and it is almost impossible to see a 3rd maximum in the morning hours (refers to Fig 1,2,3). In best case there a weak enhancement best visible in autumn. Therefore, I recommend not to call it "peak" in the text.

**Response:**

We agreed with the referee. We have made the changes as per your suggestion.

4. I have one question concerning Fig.6. Is there a special reason why you choose the ftEs parameter? Do other parameters like fbEs or foEs show similar results?

**Response:**

Thank the referee for pointing out this question. The top frequency (ftEs) is obtained in ionosonde as the foEs, meaning the maximum frequency of the Es layer. We called the ftEs since we are not distinguishing between ordinary and extraordinary traces in the data. In other words, it is just a matter of nomenclature. Also, we do not choose the fbEs because Palmas is a station near the magnetic equator, and we could have layers of irregularities that would not block the upper regions. However, we also did an analysis with fbEs, and the peaks coincide with ftEs (with a slightly less expressive value). Therefore, there would be no substantial changes using ftEs. We include this explanation in lines 363 to 377.

5. In Fig. 8, you present model results showing the electron concentration over the course of the day. When inspecting the right hand side plots, I see a large discrepancy to your Es observations from the ionosonde that I don't understand. E.g., during December conditions (upper right plot) there are two obvious ion concentration modes travelling downward. These two modes appear slightly higher and steeper compared to the upper left plot containing the dirunal and semidiurnal tidal component only which coincides with the Esh observations in Fig 2, 3. But: Especially for December the morning mode of observed Esh in Fig 3 is much stronger compared to the afternoon mode. This is totally opposite to the model outputs of electron concentration. Is this a problem from the model? Or is it a problem in the determination of the Es type? Please explain this contrary behaviour.

**Response:**

The reviewer is correct. This is a limitation in the model because we used the observation winds with extrapolation that we explain better in lines 143 to 162. However, we include the model results because we intend to show that the terdiurnal tidal mode can influence the Es layer. The simulations show that the Es layer`s electron density increases when we include the terdiurnal tide. To better see this behavior and to compare with the observational data, we have added a table where we show a clear comparison of the densities of the Es layers simulated in MIRE with the D+S and D+S+T components and the maximum daily average density for those months found in the ionosonde data. In the last column of Table 2, we also included the maximum average daily densities for the months that had the most pronounced peak. Notice that the observational data agree more with the simulations, including the terdiurnal mode. Additionally, we discussed these results in this new version (lines 363 to 377).

| Month | Electron Density Peak D+S (MIRE) | Electron Density Peak D+S+T (MIRE) | Electron Density Peak (Ionosonde and MIRE months) | Electron Density Peak (Ionosonde – Most pronounced peak) |
|---|---|---|---|---|
| December | 5.09 | 5.81 | 5.77 (Dec) | 5.51 (Feb) |
| April | 4.93 | 5.22 | 5.43 (Apr) | 5.43 (Apr) |
| July | 4.50 | 4.92 | 5.33 (Jul) | 5.33 (Jul) |
| October | 4.92 | 5.3 | 5.66 (Oct) | 5.37 (Set) |

6. Please let the reader know in the text that you seasons refer to Southern Hemispheric conditions (sorry in case I missed it) only in order to avoid any misunderstanding.

**Response:**

Ok. We make this clear now with the southern latitude expressed on line 96 and the Southern Hemisphere on line 202.

Finally, we would like to thank Referee 2 for the previous suggestions and corrections to improve the manuscript.

---

## Referee Report (RR1)

Sporadic-E layers (Es) are thin and dense layers in the ionospheric E region. Es can significantly affect the propagation of radio waves in the ionosphere, therefore has an important impact on the radar, satellite communication, and navigation. This manuscript investigate the terdiurnal tidal periodicity in the Es layer and a simulation is conducted to study the terdiurnal tidal effect on the Es formation. I have some points to help the discussions. Thus, I invite the authors to clarify before the publication.

1 I confuse about the statement "In summer and autumn, we see three well-defined peaks in a superimposed summation of the Es layer types per hour. We also observed that the modulation of the terdiurnal tide on the Es occurrence rates minimizes in December, the beginning of the summer season " in the abstract part.

2 The information about how different types of Es layer are defined should be provided in the Methodology part. And the method of calculating Es occurrence rate should be added.

3 Line 175: The description about Figure 1 is not accurate. For instance, I only see the peak of the Es occurrence rate at 01-02 UT UT during Autumn. Besides, the authors state "a sharp decrease occurs near dawn at 08 UT", however, I only notice that there was a drop at 07-08 UT.

4 Line 185: I cannot see a clear 8-h periodicities in Figure 1 (a), I will recommend that the author can add more data to conduct this statistical analysis.

5 The label of x-axis is a little confusing.

6 Why the authors conclude that "The slight increase in the occurrence rate between around 03-04 UT during the spring season might suggest that besides the dominant terdiurnal tidal periodicities, there was also a weaker quarterdiurnal (6-h) oscillation affecting the *Es* layer development"

7 Lines 255-260: Why the authors make this conclusion "This is probably related to the tendency for the amplitude of the migrating terdiurnal tide with zonal wavenumber 3 (TW3) to increase, generally from January to March 260 within $\pm$ 10° of latitude."

---

## Referee Report (RR2)

Comments:

1. The migrating terdiurnal tide with zonal wavenumber 3 (TW3) in Es layer has been identified by Tang et al. (2022, JGR).

2. The label of Figure 5 is not clear.

---

## Editor Decision (ED1)

Dear Dr. Fonte, dear co-authors,

Thank you for the substantive answers to the opponents' comments and for the modifications made to the text in accordance with the recommendations of both referees. Recently the manuscript has undergone the second revision and one of the referees come back to us with some specific comments (please, see below). Please, consider the comments and send us back your response and the revised version of the manuscript with indicated changes. At the current stage the manuscript still needs revision.

Kindest regards

Yours cordially

D. Buresova

Sporadic-E layers (Es) are thin and dense layers in the ionospheric E region. Es can significantly affect the propagation of radio waves in the ionosphere, therefore has an important impact on the radar, satellite communication, and navigation. This manuscript investigates the terdiurnal tidal periodicity in the Es layer and a simulation is conducted to study the terdiurnal tidal effect on the Es formation.

I have some points to help the discussions. Thus, I invite the authors to clarify before the publication.

1. I confuse about the statement "In summer and autumn, we see three well-defined peaks in a superimposed summation of the Es layer types per hour. We also observed that the modulation of the terdiurnal tide on the Es occurrence rates minimizes in December, the beginning of the summer season " in the abstract part.

2. The information about how different types of Es layer are defined should be provided in the Methodology part. And the method of calculating Es occurrence rate should be added.

3. Line 175: The description about Figure 1 is not accurate. For instance, I only see the peak of the Es occurrence rate at 01-02 UT UT during Autumn. Besides, the authors state "a sharp decrease occurs near dawn at 08 UT", however, I only notice that there was a drop at 07-08 UT.

4. Line 185: I cannot see a clear 8-h periodicities in Figure 1 (a), I will recommend that the author can add more data to conduct this statistical analysis.

5. The label of x-axis is a little confusing.

6. Why the authors conclude that "The slight increase in the occurrence rate between around 03-04 UT during the spring season might suggest that besides the dominant terdiurnal tidal periodicities, there was also a weaker quarterdiurnal (6-h) oscillation affecting the Es layer development".

7. Lines 255-260: Why the authors make this conclusion "This is probably related to the tendency for the amplitude of the migrating terdiurnal tide with zonal wavenumber 3 (TW3) to increase, generally from January to March 260 within ±10° of latitude".

---

## Author Response (AR2)

Letter to the Editor

Dear Dr. Fontes, dear co-authors,

Thank you for the substantive answers to the opponents' comments and for the modifications made to the text in accordance with the recommendations of both referees. Recently the manuscript has undergone the second revision and one of the referees come back to us with some specific comments (please, see below). Please, consider the comments and send us back your response and the revised version of the manuscript with indicated changes. At the current stage the manuscript still needs revision.

Kindest regards

Yours cordially

Dra. Buresova.

*Dear Editor Dra. Buresova,*

*We are glad to send our review of the article entitled "Effects of the terdiurnal tide on the Sporadic-E layers (Es) development at low latitudes over the Brazilian sector". We included the suggestions that the Reviewer addressed.*

*Best Regards,*

*Pedro Fontes et al.*

**Responses to the Comment and/or Suggestions from Referee 3**

Sporadic-E layers (Es) are thin and dense layers in the ionospheric E region. Es can significantly affect the propagation of radio waves in the ionosphere, therefore has an important impact on the radar, satellite communication, and navigation. This manuscript investigates the terdiurnal tidal periodicity in the Es layer and a simulation is conducted to study the terdiurnal tidal effect on the Es formation.

I have some points to help the discussions. Thus, I invite the authors to clarify before the publication.

*Thank you very much for the revision given by the referee. We have carried out a revision of the manuscript considering all the referee's comments.*

**Comments:**

1. I confuse about the statement "In summer and autumn, we see three well-defined peaks in a superimposed summation of the Es layer types per hour. We also observed that the modulation of the terdiurnal tide on the Es occurrence rates minimizes in December, the beginning of the summer season" in the abstract part.

**Response to the reviewer:** Figure 3a shows the three peaks in the total percentage (black line) of summer that indicates a terdiurnal oscillation, so we mentioned that "we see three well-defined peaks in the overlapping sum of Es per hour layer types in summer". However, for the summer, specifically in December, the increases in Es occurrence rates associated with amplitude modulation due to terdiurnal tides are smaller than in the other summer months. It can be seen in Fig. 3(a) by comparing the amplitude modulation for the three months used in the summer season analysis. We changed the sentence in the abstract for a better understanding.

2. The information about how different types of *Es* layer are defined should be provided in the Methodology part. And the method of calculating *Es* occurrence rate should be added.

**Response to the reviewer:** Thank the reviewer for pointing out this fact. We added a phrase in lines 113-116 to mention that the different *Es* layer types are defined according to their format in ionograms, meaning which the physical formation mechanism is acting, as seen in the U.R.S.I. Handbook of Ionogram Interpretation and Reduction (Piggott and Rawer, 1972).

Finally, a description of periodogram analysis by the Lomb-Scrage method (Lomb, 1976; Scargle, 1982) performed with the fbEs parameter for the four seasons of the year 2008/09 was added in the last paragraph of Section 2.1 (lines 129-135).

3. Line 175: The description about Figure 1 is not accurate. For instance, I only see the peak of the Es occurrence rate at 01-02 UT UT during Autumn. Besides, the authors state "a sharp decrease occurs near dawn at 08 UT", however, I only notice that there was a drop at 07-08 UT.

**Response to the reviewer:** Thanks to the reviewer for pointing out this statement. We modified the description of Figure 1 in Sect. 3.1 to facilitate the understanding these nighttime percentages as suggested by the reviewer.

4. Line 185: I cannot see a clear 8-h periodicities in Figure 1 (a), I will recommend that the author can add more data to conduct this statistical analysis.

**Response to the reviewer:** Due to the doubt of the reviewer, we added a new technique to be clearer. Thus, in Figure 6 is shown a statistical periodogram analysis with the Lomb-Scargle method (Lomb, 1976; Scargle, 1982), also described in the methodology. This analysis clearly shows the periodicity of the terdiurnal component (8 h) in all seasons of 2008/09, in addition to the 6-hour oscillation in the autumn, winter, and spring seasons (lines 330-344).

5. The label of x-axis is a little confusing.

**Response to the reviewer:** Ok. We changed the hourly interval of the x-axis in Figure 1 to just hours (1-24 hours), standardizing with the x-axis of the other figures for easier interpretation.

6. Why the authors conclude that "The slight increase in the occurrence rate between around 03-04 UT during the spring season might suggest that besides the dominant terdiurnal tidal periodicities, there was also a weaker quarterdiurnal (6-h) oscillation affecting the Es layer development".

**Response to the reviewer:** We observed a weak 6-hour oscillation in the spring (Figure 3). This behavior is visible more clearly in Figure 6, which we added in this new version. Figure 6 shows that the quarterdiurnal (6-hour) tidal component is present in the autumn, winter, and spring seasons (lines 330-344).

7. Lines 255-260: Why the authors make this conclusion "This is probably related to the tendency for the amplitude of the migrating terdiurnal tide with zonal wavenumber 3 (TW3) to increase, generally from January to March 260 within ±10° of latitude".

**Response to the reviewer:** We suggest this behavior since we found in the literature some studies that indicate this tendency of the TW3 tide to increase its amplitude between January and March by ±10° of latitude, as described in Moudden and Forbes (2013) and Pancheva et al. (2013). In addition, there is also a study at the latitude of São João do Cariri (Guharay et al., 2013), a region near Palmas in the Brazilian sector, which shows the oscillation of the terdiurnal tide of the zonal and meridional winds are present, agreeing to the conclusions of the above authors mentioned here. We included a paragraph with these citations to the works mentioned above in lines 273-285.

**References:**

Guharay, A., Batista, P. P., Clemesha, B. R., Sarkhel, S., and Buriti, R. A.: On the variability of the terdiurnal tide over a Brazilian equatorial station using meteor radar observations, *Journal of Atmospheric and Solar-Terrestrial Physics*, 104, 87–95, https://doi.org/10.1016/j.jastp.2013.08.021, 2013.

Lomb, N. R.: Least-squares frequency analysis of unequally spaced data, *Astrophysics and Space Science*, 39, 447–462, https://doi.org/10.1007/BF00648343, 1976.

Moudden, Y., and Forbes, J. M.: A decade-long climatology of terdiurnal tides using TIMED/SABER observations, *Journal of Geophysical Research: Space Physics*, 118, 4534–4550, https://doi.org/10.1002/jgra.50273, 2013.

Pancheva, D., Mukhtarov, P., and Smith, A. K.: Climatology of the migrating terdiurnal tide (TW3) in SABER/TIMED temperatures, *Journal of Geophysical Research: Space Physics*, 118, 1755–1767. https://doi.org/10.1002/jgra.50207, 2013.

Scargle, J. D.: Studies in astronomical time series analysis. II - Statistical aspects of spectral analysis of unevenly spaced data, *The Astrophysical Journal*, *263*, 835-853, https://doi.org/10.1086/160554, 1982.

*Finally, we would like to take this opportunity to thank the reviewer for kindly evaluating our paper helping to greatly improve its quality.*

---

## Author Response (AR3)

**Responses to the Comments and/or Suggestions from Referee**

**#Topical Editor:**

Thank you for evaluating and incorporating the reviewers' comments in the text of the manuscript. I am pleased to note that after the second revision, only a small step is missing from the acceptance of your paper. Please consider the opponent's additional requirements and incorporate them into the manuscript. After this modification, I will recommend your article for publication.

*Reply:  The authors thank Editor for considering the possible recommendation of this manuscript for publication in Annales Geophysicae. The authors also thank the additional minor corrections recommended by referee to improve the present manuscript. We have carried out a revision of the manuscript considering all the referee's comments.*

**#Referee's Comments:**

The additional comments of the referee:

1.The migrating terdiurnal tide with zonal wavenumber 3 (TW3) in Es layer has been identified by Tang et al. (2022, JGR).

*Reply: Thanks for recommending this reference. We have added a sentence citing Tang et al. (2022) between lines 327-328 in Sect. 3.1, page 17.*

2.The label of Figure 5 is not clear.

*Reply: The caption of Figure 5 has been changed as recommended by the reviewer.*

**The following reference has been added to the manuscript:**

Tang, Q., Zhou, C., Liu, H., Du, Z., Liu, Y., Zhao, J., Yu, Z., Zhao, Z., and Feng, X.: Global Structure and Seasonal Variations of the Tidal Amplitude in Sporadic-E Layer, *Journal of Geophysical Research: Space Physics*, 127, 1-14, https://doi.org/10.1029/2022JA030711, 2022.

*Finally, we would like to take this opportunity to thank the reviewer and topical editor for kindly evaluating our paper helping to greatly improve its quality.*